

# A case study on the impact of severe convective storms on the water vapor mixing ratio in the lower mid-latitude stratosphere observed in 2019 over Europe

Dina Khordakova[1], Christian Rolf[1], Jens-Uwe Grooß[1], Rolf Müller[1], Paul Konopka[1], Andreas Wieser[2], Martina Krämer[1], and Martin Riese[1]

[1]Institute of Energy and Climate Research (IEK-7), Forschungszentrum Jülich, Jülich, Germany
[2]Institute of Meteorology and Climate Research, Department Troposphere Research (IMK-TRO) Karlsruhe Institute of Technology, Karlsruhe, Germany

**Correspondence:** Christian Rolf (c.rolf@fz-juelich.de)

**Abstract.** Extreme convective events in the troposphere not only have immediate impacts on the surface, they can also influence the dynamics and composition of the lower stratosphere (LS). One major impact is the moistening of the LS by overshooting convection. This effect plays a crucial role in climate feedback as small changes of water vapor in the upper troposphere and lower stratosphere (UTLS) have a large impact on the radiation budget of the atmosphere. In this case study we investigate water vapor injections into the LS by two consecutive convective events in the European mid-latitudes within the framework of the MOSES (Modular Observation Solutions for Earth Systems) measurement campaign during the early summer of 2019. Using balloon-borne instruments, rare measurements of the convective water vapor injection into the stratosphere were performed. The magnitude of the water vapor reached up to 12.1 ppmv with an estimated background value of 5 ppmv. Hence it is in the same order of magnitude as earlier reports of water vapor injection by convective overshooting above North America. However the overshooting took place in the extra-tropical stratosphere and has an impact on long-term water vapor mixing ratios in the stratosphere compared to the Monsoon-influenced region in North America. At the altitude of the measured injection a sharp drop in a local ozone enhancement peak makes the observed composition of air very unique with high ozone up to 696 ppbv and high water vapor up to 12.1 ppmv. While ERA-Interim data does not show any signal of the convective overshoot, the measured values in the LS are underestimated by MLS satellite data and overestimated by ERA5 reanalysis data. Backward trajectories of the measured injected air masses reveal that the moistening of the LS took place several hours before the balloon launch. This is in good agreement with reanalyses and satellite data showing a strong change in the structure of isotherms, and a sudden and short-lived increase in potential vorticity at the altitude of the trajectory, as well as low cloud top brightness temperatures during the overshooting event.

## 1 Introduction

Extreme weather events tend to not only have immediate consequences on nature and the built environment, but also greater long-term impacts on the climate and ecosystems. Such extreme events include long-lasting drought phases, extreme precipita-



tion, heat and cold waves (periods of extreme warm or extreme cold air or sea surface temperature), as well as unusually strong hurricanes and storms. Events that have been previously considered extreme are becoming increasingly frequent and have the tendency to become the new routine (Walsh et al., 2020). Extreme convective events in the troposphere also have an influence

on the lower stratosphere (LS). One of the impacts on the LS is the in-mixing of tropospheric air masses by overshooting convection and the coherent transport of moisture into the dry lower stratosphere. Stratospheric water vapor is determined by the entry mixing ration of $H_2O$ at the tropopause and a chemical contribution by the oxidation of $CH_4$ to $H_2O$ (Randel et al., 1998). Through the analysis of multiple data sets, the water vapor background value in the LS is found to be $\approx 5$ ppmv (Pan et al., 2000; Hegglin et al., 2009); any stronger enhancements of water vapor in the LS are likely caused by the in-mixing of

tropospheric air masses (Smith et al., 2017; Wang, 2003). Stratospheric water vapor influences the climate and the chemistry of the atmosphere and plays a significant role in the positive feedback of global climate warming (Smith et al., 2017; Dessler et al., 2013b). The feedback effect of water vapor in the stratosphere is about $0.24 \, W/m^2$ for each 1 ppmv increase assuming an equal distribution globally (Solomon et al., 2010; Forster and Shine, 1999). Even small changes of the water vapor mixing ratio in the upper troposphere/lower stratosphere (UTLS) result in large radiative effects (Solomon et al., 2010; Riese et al., 2012).

A moistening of the lowermost stratosphere also has an impact on the chemistry of this region. Water vapor is a source of $HO_x$ radicals and enhances the reactivity of stratospheric sulphate aerosol particle. Anderson et al. (2012) have hypothesized that the moistening of the lowermost stratosphere by convective overshooting can lead to severe ozone depletion in summer in the mid-latitudes through heterogeneous chlorine activation. However, in a detailed analysis of the relevant chemical processes, Robrecht et al. (2019, 2020) conclude that convective moistening only has a minor impact on stratospheric ozone and the

mid-latitude ozone column.

The contribution of overshooting convection to the moisture budget of the lower stratosphere and a potential increase of overshooting convection with global warming is still under discussion (Jensen et al., 2020).

It was shown in previous work (Smith et al., 2017; Dessler et al., 2008, 2013a, 2014) that deep convective events can penetrate the tropopause and have a significant impact on the water vapor concentration in the lower stratosphere. Jensen et al.

(2020) showed that the primary region for direct convective hydration of the extra-tropics is located over North America. These direct injections over the North American continent (NA) have been evaluated in several case studies (Weinstock et al., 2007; Homeyer and Kumjian, 2015; Homeyer et al., 2017; Smith et al., 2017) and, the long-term behavior was analyzed. In our case study we investigate the transport of water vapor into the extra-tropical lower stratosphere injected by deep convective events over Europe using rare observations within a MOSES (Modular Observations Solutions of Earth Systems) measurement

campaign. MOSES aims to investigate extreme events across the Earth compartments in order to understand the short- and long-term influences of such events (Weber and Schuetze, 2019). In-situ measurements were made during early summer 2019 at a mid-latitude site in the eastern part of Germany. Balloon-borne light-weight instruments recorded water vapor, ozone, temperature and pressure immediately before and after a thunderstorm with strong convection that passed the measurement site. Two cases of overshooting convection on two consecutive days (10 and 11 June 2019) are discussed in this study. Both

cases show that significant amounts of water vapor can be transported into the lower stratosphere by deep convective events over central Europe and not just in the North American and Asian Monsoon region. We show that water vapor mixing ratios





in the same order of magnitude as the data recorded over NA can also be found deep in the extra-tropics over central Europe. Using back trajectories as well as ECMWF data and MLS data we analyze the entry point of the tropospheric air masses transported into the stratosphere.

Section 2 introduces the instruments and methods used while sections 3.1, 3.2, and 3.3 describe the two events and the results of the balloon profile measurements. In section 3.4 the data is compared to the data of the ECMWF ERA5 reanalysis data set, while in section 3.5 the time and location of the origin is discussed using backwards trajectories and satellite data. In section 4 the results are discussed, while section 5 concludes the outcome of the study.

## 2    Data and methods

### 2.1    Balloon measurements within MOSES

In this study we analyze data collected during a MOSES measurement campaign in 2019. The campaign took place from the middle of May to the end of July as part of a collaboration of 8 Helmholtz Association research centers. The objective of the measurement campaign was to capture extreme hydrological events throughout the different Earth compartments: atmosphere, ground and running waters. In the Eastern Ore Mountains in Germany, close to the city of Dresden, a 3-month measurement

campaign with intense operational phases (IOPs), was performed. During these IOPs the teams operated on demand to capture the development and cycle of convective events. Our main measurement site was located adjacent to Börnchen village in the low mountain range at 50.80 N and 13.80 E. The team from Forschungzentrum Jülich (FZJ) focused on small-scale deep convective events and their impact on the stratosphere. During the campaign period two of these events occurred and were observed with balloon borne measurements. Figure 1 schematically shows the measurement procedure. As a convective cell was approaching

the measurement site, two weather balloons were launched to measure the state of the atmosphere. The first balloon was launched just before the convective cell reaches the measurement site; the second balloon was launched immediately after the storm cell passed the measurement site as soon as the rain stopped.

   Two kinds of measurement balloons were used. The first version is a 200 g latex balloon equipped with a Vaisala radiosonde RS41-SGP, which recorded the location of the balloon as well as the altitude, pressure, temperature and moisture of the

atmosphere and transmitted the data to the ground station at the measurement site. The temperature sensor of the radiosonde has an uncertainty of 0.3 K below 16 km and 0.4 K above. The uncertainty of the humidity sensor is given as 3% and the pressure sensor has an uncertainty of 1.0 hPa at ambient pressure above 100 hPa; 0.3 hPa between 10 and 100 hPa and 0.04 hPa below 10 hPa. The second version is a 1500 g balloon additionally equipped with a payload carrying multiple in-situ intruments. An ECC instrument (Electrochemical Concentration Cell, Smit et al., 2007) was used to measure ozone mixing ratios which has an

uncertainty of ≈5% below 20 km (Smit et al., 2007; Thompson et al., 2019; Tarasick et al., 2021), and a CFH (Cryogenic Frost-point Hygrometer, Vömel et al., 2007) was used to measure the low water vapor concentration prevailing in the tropopause region and in the stratosphere. The uncertainty of the CFH instrument given as 4% in the troposphere and below 10% in the stratosphere. The payload also contained a Compact Optical Backscatter Aerosol Detector (COBALD) to measure backscatter from different types of particles during nighttime (Brabec et al., 2012). This is referred to as 'large payload' in the following.





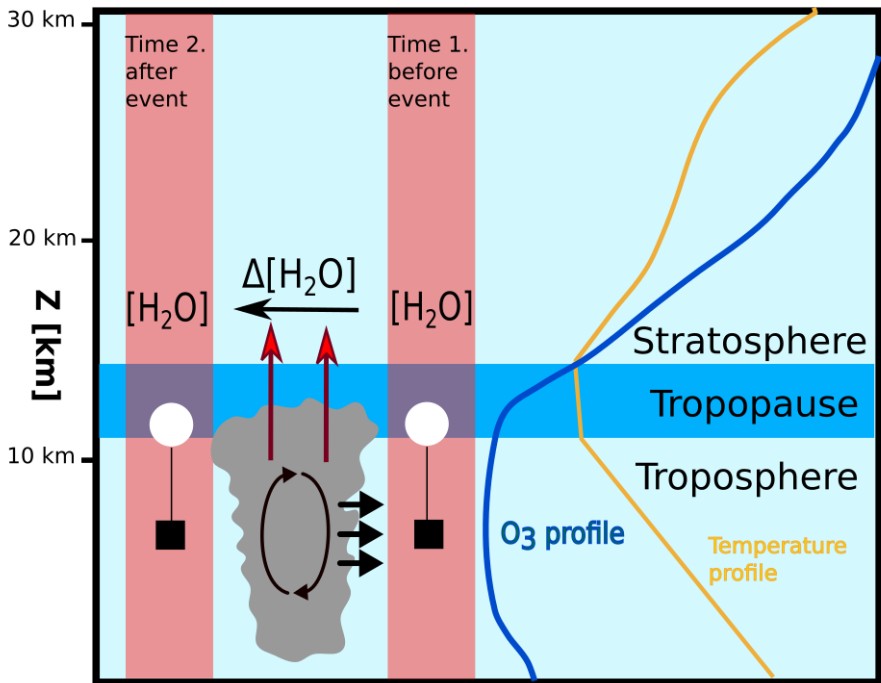

**Figure 1.** Schematic of the measurement strategy. A balloon is started right before and immediatly after a deep convective event has passed the measurements site. On the right hand site the approximate ozone (blue) and temperature (yellow) climatological profiles are shown. The amount of water vapor transported into the stratosphere is investigated by the difference between the two profiles above the lapse rate tropopause according to the WMO definition.

However, the measurements taken by the COBALD instrument were not used for the analysis presented here. A picture of the entire payload with radiosonde, ECC, CFH and COBALD is shown in Figure A1. The payload is adapted from the setup used by the GRUAN (The Global Climate Observing System (GCOS) Reference Upper-Air Network) set-up (Dirksen et al., 2014). A more detailed description of the instruments can be found in the appendix A.

Figure 2 displays all available water vapor profiles (18 profiles available) measured in mid-latitudes by the authors with the
RS41 and the CFH as a reference instrument (see appendix A2) between 2018 and 2020. When considering data up to 20 km the correlation of the RS41 and CFH data is 0.975. However, the deviation between both instruments increases above ≈20 km and the correlation of data is reduced to 0.45 for data points measured between 20 km and 30 km. In the mid-stratosphere the low humidity in combination with low pressure does not allow the RS41 to conduct reliable measurements. Therefore, these astonishing good results offer reliability using accurate RS41 humidity measurements up to a height of 20 km which are used
in this study in the case of flights without the CFH instrument.

During the first event, on the 10 June 2019, we launched 1 radiosonde and 1 large payload, while only 5 radiosondes were used during the second event, on the 11 June 2019 due to logistical reasons. In most cases, the balloons reached far into the

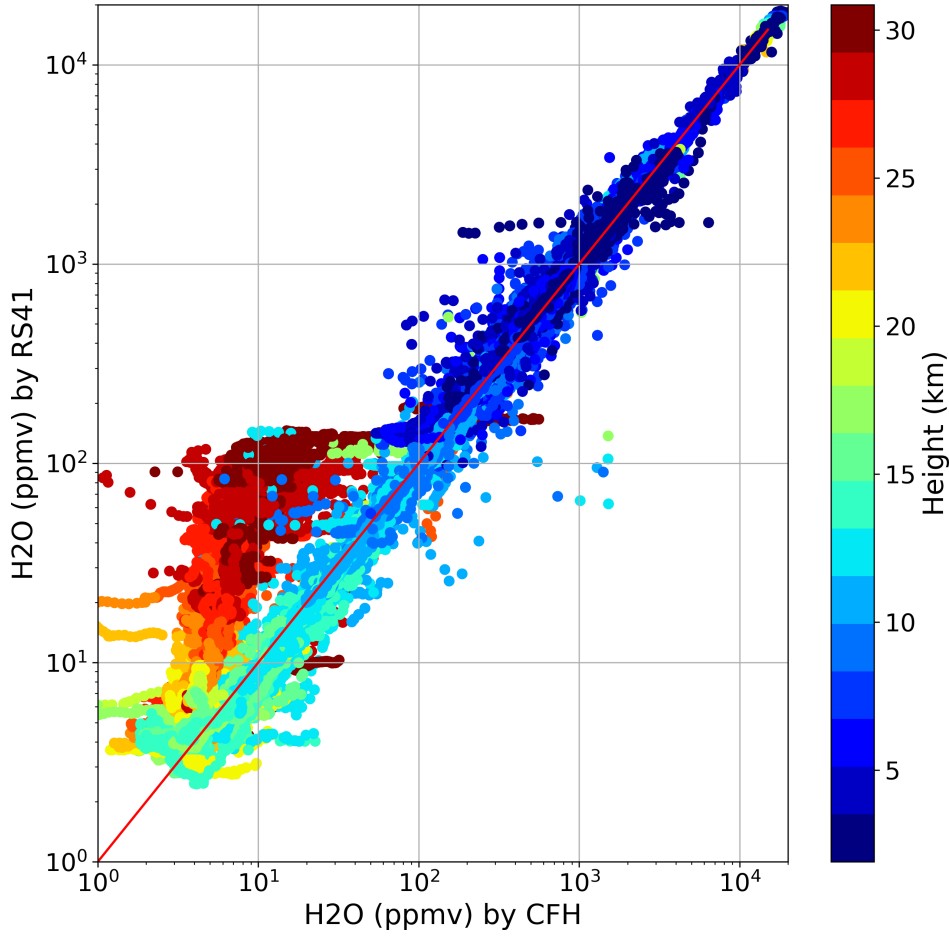

**Figure 2.** Correlation of water vapor mixing ratios measured simultaneously by RS41 and the CFH. The color code represents the altitude of the measurement. All data measured by the authors between 2018 and 2020 are used.

stratosphere at altitudes of up to 22 km with radiosondes only and up to 35 km with larger balloons, which were equipped with the above-mentioned instruments and captured the entire UTLS region during ascent and descent. During the first balloon

launch of the first convective event the connection to the radiosonde was lost for about 20 minutes, and the data between 11 km and 18 km altitude were lost during the ascent, but all other sounding data is complete.

## 2.2 Aura Microwave Limb Sounder (MLS)

The Microwave Limb Sounder is a local instrument operating on the Aura satellite. The sun-synchronous polar orbit satellite has an inclination of 98° and an equator-crossing time of 13:45 UTC ±15 min. It was launched on the 15 July 2004 and has

been operating ever since. The measurements are in limb-viewing geometry on the A-Train orbit and are in the spectral range of thermal emission, thus day- and night-time measurements are available. The temperature and pressure is retrieved from the





118 GHz band, water vapor from the 190 GHz band and ozone and CO from the 240 GHz band (Schoeberl et al., 2006; Waters et al., 2006). In this work, data from the version 4 retrieval algorithm was used to obtain the data presented. In this study we show ozone, water vapor and CO mixing ratios. MLS Version 4 data is provided on 36 different pressure levels ranging from
316 hPa to 0.002 hPa as described in Livesey et al. (2017) and the data quality is described in Pumphrey et al. (2011). One of the main improvements of version 4 is improved cloud detection, excluding cloudy radiances causing corrupted profiles. This improvement increases the quality of our data set as our area of interest is covered with clouds.

## 2.3   ECMWF ERA5

The European Centre for Medium-Range Weather Forecasts (ECMWF) produces numerical weather forecasts and provides a
meteorological data archive. In this study, we use the ERA5 data set which is a global reanalysis covering the period from 1979 until present (Hersbach et al., 2020). The spatial resolution is about 30 km and contains 137 vertical levels from the surface to an altitude of 80 km. In this work the ERA5 data from May to June 2019 was used with hourly temporal resolution. The data set was interpolated to isentropic levels and potential vorticity (PV) was added on the individual isentropic levels (Ertel, 1942). In the Northern Hemisphere PV values above 2 PVU are typical for the stratosphere while values below 2 PVU are typical for
the troposphere, where $1 \mathrm{PVU} = 1 \cdot 10^{-6} \mathrm{km}^2 \cdot \mathrm{kg}^{-1} \cdot \mathrm{s}^{-1}$ (Kunz et al., 2011). Additionally, we calculated the vertical gradient of potential temperature, which is part of the PV definition, defined as:

$$PV = -g \cdot (\zeta + f) \cdot \frac{\partial \theta}{\partial p} \tag{1}$$

where $g$ is the gravitational acceleration, $\zeta$ the relative isentropic vorticity, $f$ the Coriolis parameter, $\theta$ is the potetial temperature and $p$ is pressure. The vertical gradient of potential temperature is $\frac{\partial \theta}{\partial p}$ and is hereafter referred to as dTheta. dTheta is negative by
definition, as with decreasing pressure the potential temperature increases in a stable atmosphere. In the troposphere, potential temperature only shows a slight increase and can be considered constant relative to the steep increase that occurs above the tropopause.

## 2.4   Trajectory calculation

In order to calculate backward and forward trajectories of the measured air masses the trajectory module of the three-dimensional
chemistry transport model CLaMS (Chemical Lagrangian Model for the Stratosphere; McKenna et al. (2002)) was used. The trajectories were initialized at pressure levels between 135 hPa and 175 hPa in steps of 5 hPa which encompasses the pressure level of the maximum water vapor enhancement measured for both cases using the same method as described in Rolf et al. (2018). Each trajectory was calculated both for 100 hours backward and forward in time. The trajectory calculation with CLaMS is based on the ERA5 horizontal wind fields and diabatic heating rates with an hourly output. In addition, temperature,
pressure, PV, water vapor and ozone mixing ratios as well as convective available potential energy (CAPE) are interpolated from ERA5 data onto the coordinate of the trajectories.





## 3 Measurement results and analysis

### 3.1 Meteorological situation at the time of the case study

From 10 to 12 June 2019 multiple severe convective storms developed over Germany. During these events, hail with a diameter
of up to 6 cm was observed and heavy rain with a daily amount of 100 mm was measured. Wilhelm et al. (2020) describe this
series of convective storms in detail. In our study, we define the events that precede the measurements taken on the evening of
10 June 2019 as Case 1 and the ones preceding the measurements taken in the evening of 11 June 2019 as Case 2. The storm of
Case 1 passed the measurement site at approximately 20:00 UTC on 10 June 2019. On the previous day, a low pressure system
with warm and humid air was brought to Central Europe while a strong wind shear caused by a lee depression was located
over the Czech Republic. A first convective storm developed in the north-eastern part of Italy and progressed westwards until it
started dissipating at around 08:00 UTC over the north-eastern part of Italy. Later in the day, in combination with strong solar
radiation, these storm precursors caused the first significant convective cell over Memmingen (southern Germany) at around
16:00 UTC. This cell developed into a super cell and caused severe damage in northern Munich at around 17:45 UTC. Multiple
super cells subsequently formed, combined over eastern Germany, and later moved towards Poland and the Baltic Sea. The
formation of super cells passed the measurement site in the Eastern Ore Mountains and balloon profiles were taken before
and after the storm cell passed. The first balloon was launched at approximately 18:00 UTC (hereinafter referred to as "profile
before") equipped only with a radiosonde. The second balloon launch, with a large balloon payload, took place at 01:00 UTC
the next day, shortly after the thunderstorm passed.

On 11 June 2019, the already warm and humid air mass was heated up to 33 °C at ground level in the afternoon. At 12:00 UTC
a first convective cell developed over the Slovenian/Austrian border and further developed over the next 7 hours to a mesoscale
convective system (MCS) covering almost all of Austria and Slovenia. With an offset of approximately 1 hour another convec-
tive cell emerged over the center of northern Italy and multiple smaller cells developed over the German-Czech border starting
at around 15:00 UTC. All these convective cells increased spatially throughout the day and unified to an MCS covering the
entirety of eastern Germany. At around 17:39 UTC a first cell developed between Dresden and Bautzen. Hail with particles
reaching a diameter of up to 4 cm was observed. This event, subsequently referred to as Case 2, was captured only with ra-
diosondes that were launched every 3 hours starting from 13:00 UTC until midnight when the last radiosonde was launched
after the storm had passed the measurement site.

### 3.2 Water vapor injection captured by balloon profiles

The measurement results of Case 1 and Case 2 can be seen in Figures 3 and 4. The measurements before and after the respective
extreme convective event (hereinafter referred to as "the event") are displayed with ascending and descending profiles, where
available. The UTLS intercept is shown with pressure levels between 240 hPa and 90 hPa and potential temperatures range
between 320 K and 420 K. A sharp transition from the characteristics of tropospheric to stratospheric air masses is clearly
discernible in all figures. The lapse rate tropopause (LRT), as defined by the World Meteorological Organization (WMO), is
at 203 hPa before and at 196 hPa pressure level after the convective event for Case 1 (see Figure 3) and at 194 hPa before and


at 200 hPa pressure level after the event for Case 2 (see Figure 4). In all cases, the cold point tropopause (CPT) is slightly (4-20 hPa) above the LRT. For Case 1, the sharp transition from the troposphere to the stratosphere is discernible by a distinct change in the course of the temperature. Additionally, an abrupt increase in ozone and a decrease of the water vapor mixing ratio towards the stratospheric background level below 5 ppmv show the difference between the two regimes. Between pressure levels of 180 hPa and 160 hPa, which correspond to potential temperature levels of 340 K and 360 K, the water vapor mixing

ratio fluctuates between 5 ppmv and 7.5 ppmv and between 6 ppmv and 14 ppmv as measured by the radiosonde and the CFH respectively, before it attains the stratospheric background value of $\approx$ 5 ppmv, which is reached within all Case 1 profiles below the 160 hPa/360 K level. It is therefore set as a background value and agrees well with results from previous studies (Pan et al., 2000). The ascent profile measured after the event shows a strong increase in water vapor measured by the radiosonde as well as by the CFH above the level of 155 hPa/365 K. The maximum value measured by the RS41 is 7.0 ($\pm$ 10%) ppmv and the

maximum value measured by the CFH is 8.6 ($\pm$ 6%) ppmv. The lower value of the RS41 results from a lagging response time of the radiosonde at higher altitudes as described in A3. Above the water vapor enhancement, at 143 hPa/375 K, the mixing ratio decreases rapidly again to the background value below 5 ppmv. This peak is only apparent in the ascending profile of the measurement. The descending profile shows no peak signatures either in the CFH or in the RS41 water vapor measurements. As the horizontal distance between the location during ascent and descent at this altitude is only 60 km and about 2 hours

(00:59 UTC/02:49 UTC) time difference, the enhancement in the water vapor mixing ratio is a localized feature. In Figures 3a and b, the vertical extent of the discussed water vapor peak are framed with a gray background. At a similar level at which the water vapor peak is located, a striking peak in the ozone profile can be seen. With a lower edge at 162 hPa/359 K and an upper edge at 145 hPa/373 K the ozone peak starts at a lower level compared to the water vapor enhancement, but is limited by the same upper edge. The steep decrease of the ozone mixing ratio occurs very sharply at the same potential temperautre level as

the sudden appearance of the water vapor peak. This is a major indicator of the in-mixing of tropospheric air into this level, which has a low concentration of ozone and a high amount of water vapor. Figure 3b clearly demonstrates that the air mass with increased water vapor also has diluted mixing ratios of ozone as the ozone mixing ratio decreases sharply at the same level as the strong increase in water vapor appears. Further evidence of the tropospheric origin of the air mass can be seen when considering the temperature profile. The temperatures typically increase with altitude throughout the stratosphere. In the

measured ascent profile, after the temperature dropped to 207.9 K at the CPT, it increases until it reaches 220.4 K at a potential temperature level of 365 K where it declines sharply to 218.63 K. In Figure 3b it becomes evident that the larger temperature dip of $\approx$ 2 K occurs suddenly at 365 K, the same level as the strong decrease within the ozone peak. The temperature drop within the water vapor enhancement might be a result of the adiabatic cooling within the overshooting top of the convection. Case 2 presents a slightly different background atmosphere as Case 1. The transition from the tropospheric to the stratospheric

regime proceeds less abruptly as can be depicted from Figure 4a and b (the water vapor mixing ratios of the flight launched at 13:14 UTC with the RS41 was corrected for an off-set bias). The CPT is further above the LRT and the CPT temperature minimum is less distinct. For Case 2 multiple background profiles exist, launched throughout the day, before the occurrence of the convective event in the night. To simplify the figure, only the tropopause of the last profile before the event is shown in Figures 4a and b. The water vapor profile shows a similar feature as Case 1. As the water vapor mixing ratio converges to




the background value, it is first disrupted by a peak reaching a value of 6.5 ppmv and returning to the background value at a
pressure/potential temperature level of 153 hPa/365 K, respectively. At this elevation a second peak is discernible with water
vapor mixing ratios of 12.1 ppmv (± 10%) at 143 hPa/371 K. As multiple balloon launches were performed throughout the
day, an increase of background water vapor mixing ratios with progressing launch time is evident. Balloon profiles launched at
19:00 UTC and 22:00 UTC show a slight water vapor enhancement up to 5.5 ppmv (±10%) at the same level as the main peak

measured in the ascending profile after the event. The descending profile also shows an increase in water vapor mixing ratio at
the same pressure altitude as the ascending profile. However this peak is wider and only about half the amplitude. Similar to
Case 1 the temperature measured during ascent shows a sharp decrease of 2 K at the potential temperature level of the highest
water vapor mixing ratio value of the peak. In Case 2 the water vapor peak is more spiked compared to the rectangular profile
visible in Case 1 (shown in Figure 3a and b). It is of further interest that all temperature profiles measured on 11 June 2019

clearly show a second tropopause at about 110 hPa while the temperature profiles of Case 1, measured only a couple of hours
before do not show such a structure.

### 3.3 Source of the ozone peak at 150 hPa

Figure 3a shows the profile measured after the event of Case 1. A strong ozone peak with values of up to 696 ppbv can be
seen starting somewhat below the water vapor peak at a pressure level of 150 hPa. Usually, it is expected to find a negative

correlation between water vapor and ozone when tropospheric air masses are injected by overshooting convection into the
stratosphere. It is therefore unexpected to find such a strong increase (about 300 ppbv) in ozone at the same level as the water
vapor injection. Figure 3b shows that a steep decrease of ozone mixing ratios at potential temperature levels between 365 K
and 375 K. This indicates a dilution of the ozone-rich stratospheric air with ozone-poor tropospheric air. However, the origin
of the ozone peak between the 160 K and 375 K potential temperature level, has to be clarified. Multiple possible explanations

might be considered. One suggestion linked strong increase in ozone to the injection of increased $NO_x$ produced by severe
lightning (Seinfeld and Pandis, 1998; Bond et al., 2001; Cooray et al., 2009). $NO_x$ is controlling the $O_3$ concentration in the
troposphere and is mainly responsible for the development of photochemical smog in the troposphere. However, the increase
of ≈300 ppbv can not be explained by that because model simulations show that the potential increase due to $NO_x$ would be
in the order of 10 ppbv (DeCaria et al., 2005).

Another ozone source can be direct corona discharge during lightning leading to ozone formation (Minschwaner et al., 2008;
Bozem et al., 2014; Kotsakis et al., 2017). However, the enhancement of ozone due to this process is reported to be in the order
of about 50 ppbv and thus can not explain the increase in the observed range. Figure 5a shows the ozone profile measured
after the event during Case 1 in comparison to the mean of all ozone profiles (8 available profiles) obtained by the authors
with multiple balloon measurements in Germany during spring and summertime measurements between 2018 and 2020. It is

evident that although the profile of Case 1 clearly exceeds the mean ozone profile at this altitude, but it remains within the usual
range of observations. Figure 5b displays the $H_2O$ - $O_3$ distribution of the same data as in the left panel. Here, the measurement
data (red dotted line) with the high amount of ozone and water vapor diverge prominently from the typical L-shaped data set
(gray dotted data), which marks the tropospheric and stratospheric regimes. It is not unusual that vertically thin filaments of





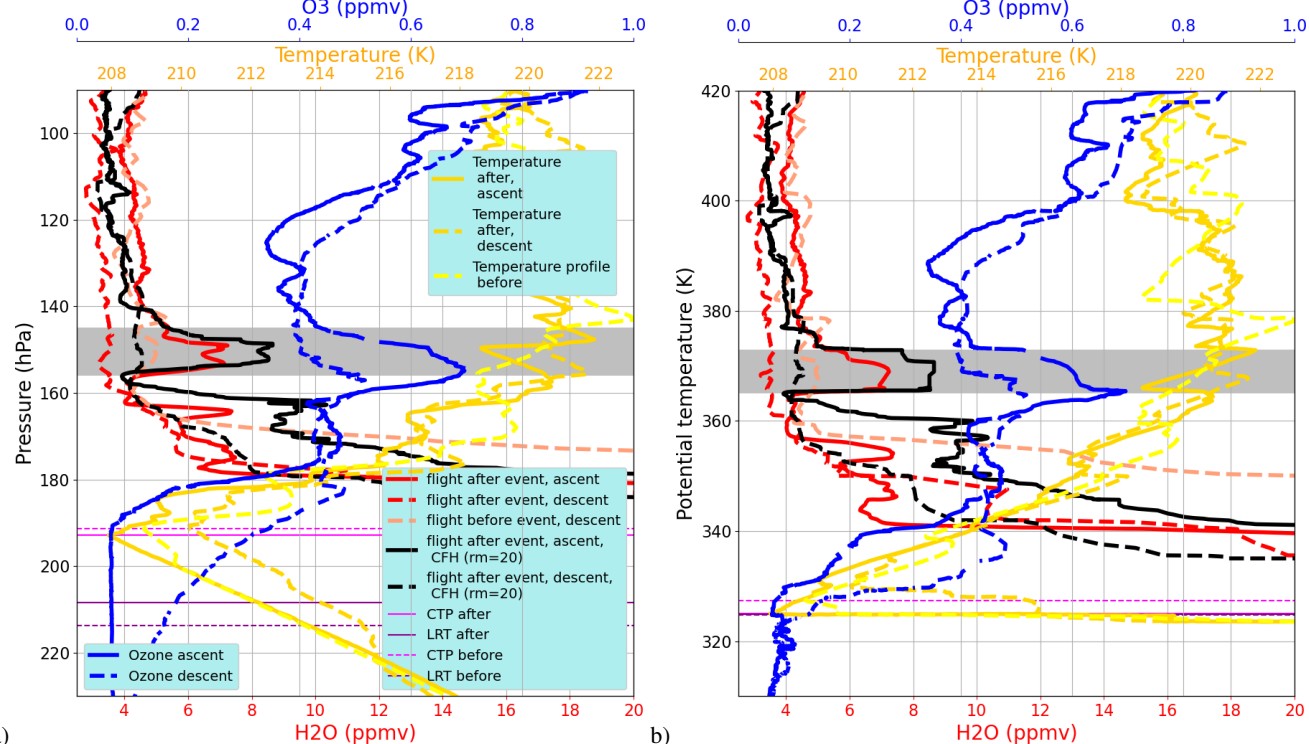

**Figure 3.** Profiles measured immediately before and after the convective event (Case 1) at 18:00 UTC on 10 June 2019 and 00:00 UTC on 11 June 2019 in the UTLS region. The water vapor measurements are shown in reddish colors for the RS41 and in black for the CFH instrument. Ozone measurements are depicted in blue. Temperature is shown in yellowish colors. a) Pressure is used as vertical coordinate. The gray region denotes the area between 145 hPa and 165 hPa in which the water vapor enhancement is observed. b) Potential temperature is used as a vertical coordinate. The different tropopauses (LRT, CPT) are shown as horizontal lines. The gray region marks the level between 365 K and 370 K in which the water vapor enhancement is observed.

ozone-rich stratospheric air masses are transported horizontally causing local ozone enhancements in vertical profiles. A model
run with CLaMS using different ECMWF data sets as input, shows an enhancement of ozone between 100 hPa and 200 hPa (not shown). This indicates the horizontal transport of ozone-rich stratospheric air, as the CLaMS model does not account for overshooting events. Figure 6 presents the ECMWF ERA5 reanalysis data at the time and approximate altitude of the observed ozone peak. A narrow ozone-rich filament extends eastward from air masses with stratospheric origin towards the measurement location. Hence, there is strong evidence that the ozone-rich stratospheric filament was transported horizontally to the location
where water vapor was injected by overshooting convection into the lowermost stratosphere. The location and development of the overshooting convection is discussed in sections 3.4 and 3.5.



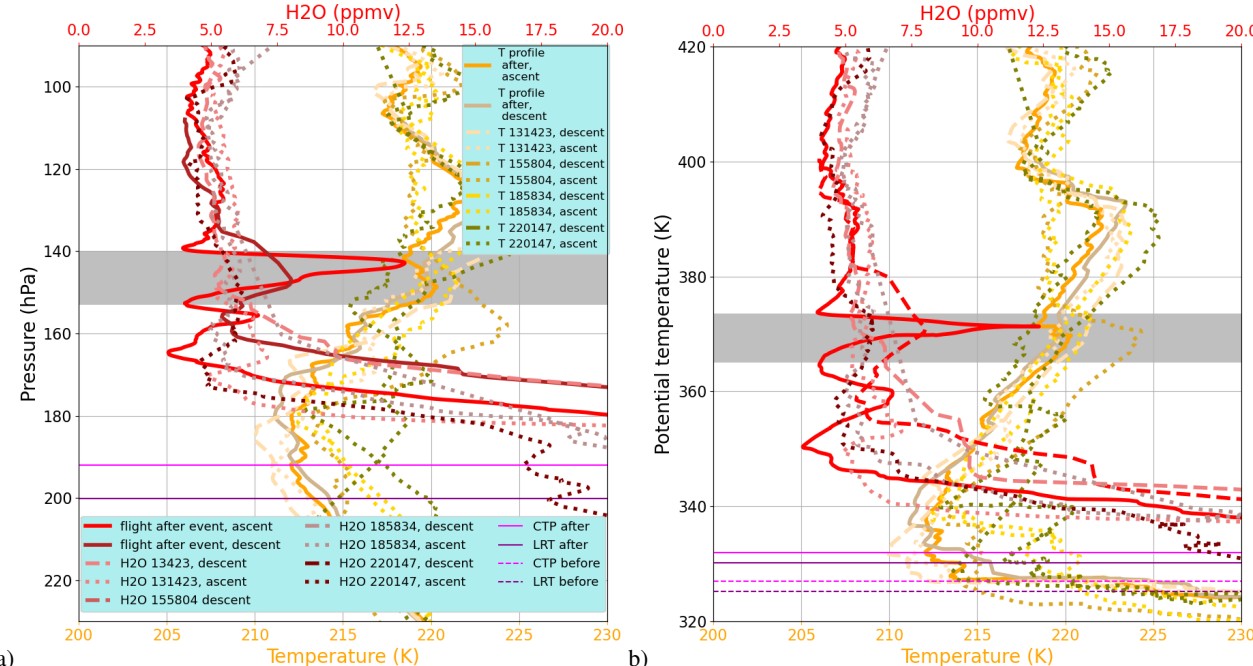

**Figure 4.** Same as Fig. 3 but for profiles measured immediatly before and after the convective event on 11 June 2019 (Case 2) passed the measurement site in the UTLS region.The water vapor mixing ratios is shown in red, measured by the radiosonde before and after the event. a) Pressure is used as a vertical coordinate. b) Potential temperature is used as a vertical coordinate.

## 3.4    Comparison to ECMWF ERA5 data

ECMWF ERA5 data is used to set the measured data in a wider context and to evaluate the events. While ERA-Interim data does not show any local signatures of the measured convection, ERA5 data reveals the signature of a convective overshooting

with multiple parameters. Here we consider CAPE, PV, potential temperature, and water vapor mixing ratio with data starting at midnight on 10 June 2019 until midnight on 14 June 2019. CAPE is the integrated amount of energy that the upward buoyancy force would act on a air parcel if it moves vertically. High CAPE values above 1000 J/kg show an increased probability of strong convective storm development in case that convection is initiated. Figure 7 panels a-c and 8 panels a-c display the distribution of CAPE at three chosen points in time across central Europe for Case 1 and 2 respectively. The white line marks the backward

and forwards trajectories which were initiated at the time and location of the measured water vapor peak for Case 1 (discussed in Section 3.5) and the black dot marks the location of the sampled air mass at the given time point according to the calculated trajectories. Very high CAPE values at the coast of Slovenia/Croatia as well as the east coast of Italy and northern Italy are evident in all chosen time frames.

Figure 7 panels a-b show that the air mass measured after the event of Case 1 is located just above a strong maximum in

CAPE over north Italy on the morning of 10 June 2019. Throughout the day, the air parcel moves close into regions of enhanced CAPE on multiple occasions along the way to the measurement site, finally reaching the center of a region with high CAPE





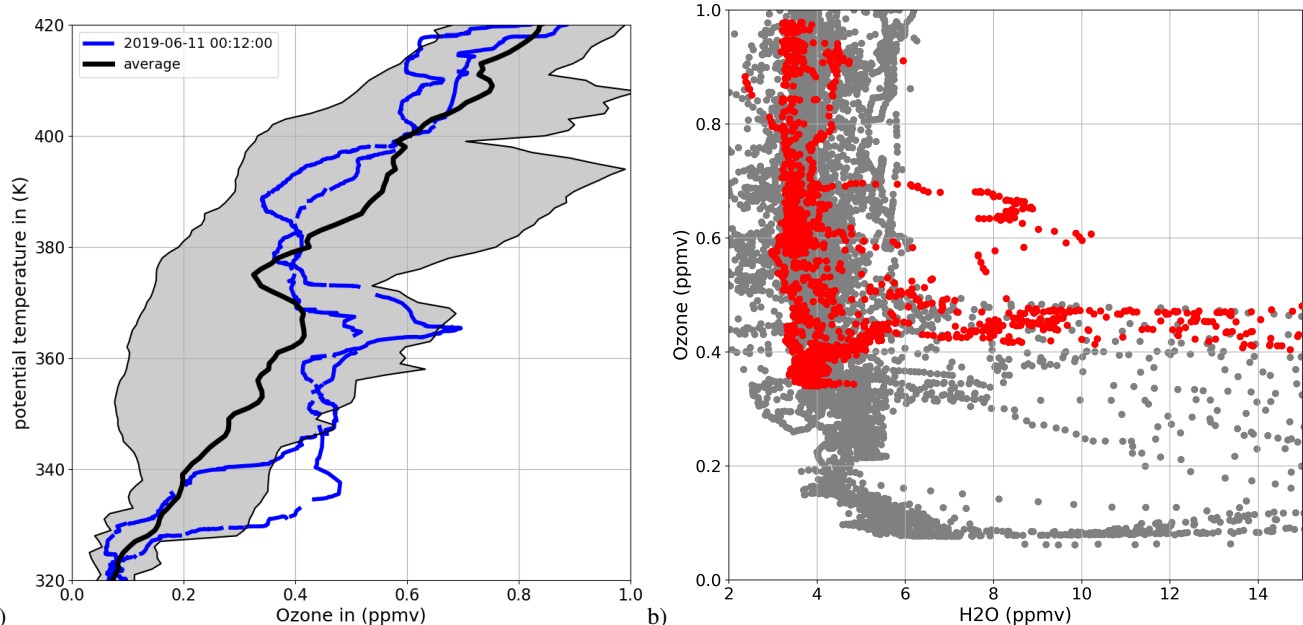

**Figure 5.** Climatology of 8 ozone profiles measured during spring and summer 2018-2020 a) Ozone profiles within the UTLS altitude range from 2018 to 2020 by the authors in potential temperature coordinates. The shaded area marks the measured range of ozone mixing ratios. b) Tracer-tracer correlation of water vapor and ozone mixing ratios within the UTLS altitude range from 2018 to 2020 by the authors in mid-latitudes.

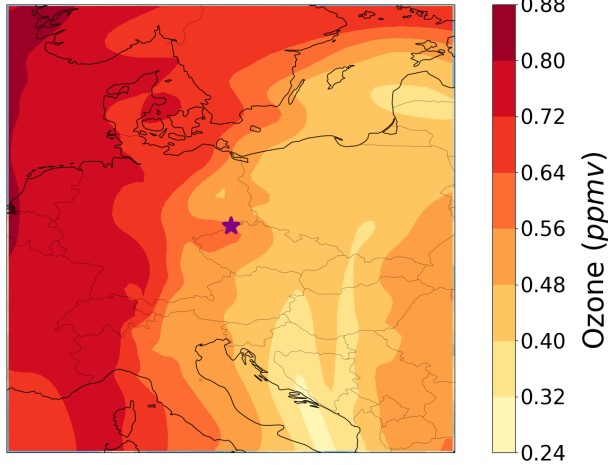

**Figure 6.** Horizontal map of ECMWF ERA5 ozone mixing ratio at a pressure level of 148 hPa on 11 June 2019 at 01:00 UTC (Case 1). The measurement site is denoted by a purple star.

close to the measurement site (Figure 7, panel c). Figures 8 panels a-c depict the same scenario for Case 2. Here, the air masses





crosses a location with high CAPE for the first time at 06:00 UTC on 11 June 2019 over Slovenia. It remains within the region of high CAPE until 13:00 UTC before it crosses the measurement site at midnight.

In contrast to the persistent and horizontally wide-ranged distribution of elevated CAPE values, a different structural evolution is observed in the potential vorticity (PV) (not shown) and dTheta. When considering dTheta at the altitude of the measured air parcel with enhanced water vapor, a strong minimum can be seen coinciding with the signature of PV for Case 1 and Case 2. For Case 1, a day before the measured event no profound signature in the dTheta structure is seen until 09:00 UTC on the morning of 10 June 2019 (Figure 7d). However at 10:00 UTC (Figure 7e) a spot signature in dTheta is apparent leading

to PV values of up to 25 PVU in the region of high CAPE values over northern Italy. This is more than twice as high as the surrounding PV values. The air mass later sampled is located at the edge of the strong dTheta enhancement with still strong values throughout the next 10 hours. This signature subsequently weakens (Figure 7e) but reappears with increased intensity (Figure 7f) and moves northwards until it dissolves at midnight. The trajectory of the air parcel moves only slightly westwards of this structure but remains inside the enhancement of PV over the entire time, although never in the center.

A similar course of events can be observed for Case 2, as shown in Figure 8. In comparison to Case 1, the trajectory of the air mass measured in Case 2 approaches further from the south. In the early morning of 11 June 2019 no significant structure or signal can be seen in the area of interest (Figure 8d). At 10:00 UTC a dipole structure in dTheta appears leading to PV values of up to 30 PVU (Figure 8e). Similar to Case 1, the signal does not develop gradually nor is it transported horizontally into the considered area, instead emerging on a very short timescale. The anomaly appears over Austria, northern Italy and over the

Czech Republic and therefore has 3 central points. The enhancement over the Czech Republic dissolves in the following hour while the other two increase in strength over the next few hours. However, all three centers dissolve until midnight when the air parcel reaches the measurement site. In Case 2 the air parcel is also constantly in the vicinity of at least one of the peaks in dTheta, but never enters areas of the extraordinary high values.

This signature in dTheta can be explained with the displacement of the isentropes upward by strong updraft winds and local

diabatic heating and thus cause an increase in the gradient of potential temperature (Qu et al., 2020). In both cases considered here, the map of the dTheta is homogeneous before convection appears until 09:00 UTC (Figure 7 panels g-i). However only one hour later, a spot signal with values of up to -2.7 K · hPa$^{-1}$ appears which is more than 3 times higher than the surrounding values. In both cases, the peak in dTheta moves along the PV enhancement and also dissolves at the same time.

Furthermore, the specific humidity in the ECMWF data was analyzed. Figure 7 panels g-i show the specific humidity of the

ERA5 data for Case 1. Figure 7g shows the hour before the first appearance of the signature of the convective storm for Case 1 at 09:00 UTC on 10 June 2019. Two peaks can be seen on the map but not close to the path of the air mass. One hour later, at 10:00 UTC, a water vapor peak emerges in the vicinity of the air mass (Figure 7h) at the same location as the enhancement in PV and dTheta. For Case 2 a similar picture is seen. While at 09:00 UTC no local enhancements in water vapor mixing ratios can be seen in the considered area, only one hour later, at 10:00 UTC a strong enhancement in water vapor is evident in the

vicinity of the considered air mass. This signature of the local enhancement is almost twice as high as the peak seen for Case 1. Similarly to Case 1, the enhancement is transported along towards the measurement site throughout the day and remains close to the measured air mass.



**Figure 7.** ERA5 data of CAPE (a-c) vertical gradient of potential temperature (dTheta, d-f), and specific humidity (g-i) at three chosen time points for Case 1 (09:0 UTC and 10:00, UTC on 10 June 2019 and 21:00 UTC). dTheta and specific humidity are displayed at a pressure level of 148 hPa. The horizontal black dashed line denotes the latitude of the measurement site and the purple star denotes the exact measurement location. The white line shows the trajectory of the measured air parcel as described in section 2.4. The black dot on the trajectory line represents the calculated location of the air mass at the given point in time.



**Figure 8.** The same as in Figure 7 but for Case 2 with three chosen points in time (09:00 UTC, 10:00 UTC and 18:00 UTC on 11 June 2019).





## 3.5 Origin and evolution of the water vapor enhancement along CLaMS trajectories

In order to determine the origin and evolution of the measured air masses containing the water vapor enhancement, 100 h

backward and 100 h forward trajectories were calculated for both cases, as described in section 2.4. Water vapor mixing ratios along the trajectories are shown in the upper panel of Figure 9. The backward trajectories are not shown before 06:00 UTC on 9 June 2019 as the data do not contain any relevant information related to the measurements. The upper panel also displays the water vapor mixing ratios along the trajectory and the data points measured by MLS within 5 degrees of latitude and longitude and an hour before or after the trajectory point (star symbols). The ERA5 water vapor mixing ratio shows a sharp

increase along the trajectory from values around 7 ppmv to values up to 15 ppmv ≈ 10 hours before the balloon measurement took place. Although multiple MLS data points were available, a clear increase in water vapor cannot be seen in the available data. Overall, the values measured by MLS are much lower compared to the ECMWF ERA5 water vapor values ranging from 2 ppmv to 6 ppmv along the calculated trajectory, while the ECMWF values vary between 5 ppmv and 18 ppmv. The middle panel shows the mixing ratios of ozone and water vapor as well as PV and CAPE values from ERA5 data along the trajectory.

The trajectory encounters high CAPE values shortly before a steep increase in water vapor and PV appear on 10 June 2019 around 10:00 UTC. The peak in CAPE is followed by a peak in PV almost doubling the preceding values of around 8 PVU. This peak is in good agreement with an increase in water vapor mixing ratios by 10 ppmv, which remains at the level between 12.3 ppmv and 17.5 ppmv throughout the following 4 days of the trajectory in contrast to the PV enhancement which decreases shortly before the balloon observations to a background value of 8 PVU. With a water vapor mixing ratio of 10 ppmv measured

by the CFH at the peak, the values obtained with the balloon payload are lower than the ERA5 values. The ozone mixing ratios do not show an impact by the convective event, but steadily decrease throughout the trajectory. The lower panel of Figure 9 shows MLS water vapor, ozone, and CO mean mixing ratios for the nearest MLS point for each time step within 300 km along the trajectory. Only 7 measurement points were found to match the criteria. CO as well as water vapor, acts as a tropospheric tracer with sources at the surface and background values in the stratosphere (Ricaud et al., 2007) and was considered here as a

potential additional tracer for convective overshooting. The nearest values of the individual data sets do not show any increase in relation to the proposed convective event. This emphasizes the small scale of the overshooting event and the local scale of the water vapor enhancement, as MLS only has a very coarse spatial resolution in the LS.

A similar picture appears for Case 2 shown in Figure 10. It must be noted that the scales differ in comparison to Figure 9. In Case 2, at midday on 10 June 2019 a series of peaks in CAPE emerge and persist throughout the next four days. Shortly

before the start of these variations in CAPE, a slight increase in PV is evident and PV values subsequently show enhanced values although not exceeding 9.5 PVU at a background of 7.5 PVU. Water vapor along the trajectory remains constant until midday on 11 June 2019 when it is slightly enhanced from approximately 6 ppmv to 10 ppmv shortly after CAPE and PV reach maximum values ≈11 hours before the measurement took place. Similar to Case 1, the water vapor continuously remained at the elevated mixing ratios throughout the end of the trajectory. For Case 2, only 4 MLS measurement points were found within

300 km of the trajectory. A slight enhancement of 1 ppmv in water vapor mixing ratio in the MLS data after the overshooting convection as well as a 30 ppmv increase in CO, which remains enhanced between 55 ppmv and 65 ppmv and also a slight

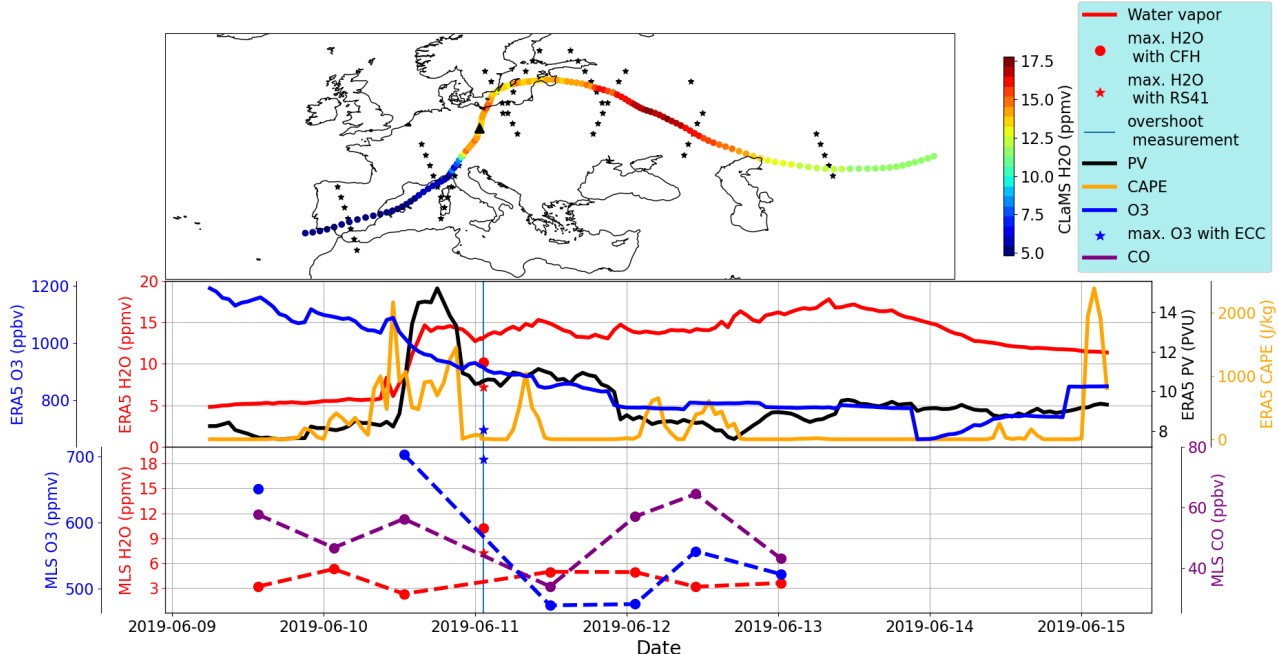

**Figure 9.** Trajectory of the measured air mass for Case 1 with MLS data taken within 5 degrees of latitude and longitude of the trajectory. The upper panel shows the trajectory on a map with the color-coded water vapor mixing ratios along the trajectory. Additionally, all MLS data points within a longitude and latitude of 5 degrees as well as within a hour before and after the individual trajectory points are shown as star symbols. In the second panel, the water vapor from ERA5 along the trajectory is displayed in red, ozone in blue, PV in black and CAPE in orange. The time of the measurement and the observed maximum water vapor mixing ratio from CFH and RS41 within the pressure levels of 145 hPa and 165 hPa are shown at the vertical blue line and with blue and red symbols, respectively. The third panel displays the same time frame with the nearest MLS measurements of ozone in blue, water vapor in red and CO in purple within a radius of 300 km of each trajectory point and an hour before or after the trajectory point.

decrease in ozone mixing ratio by 20 ppbv can be seen. Here it is emphasized that the overshooting event of Case 2 has likely a wider horizontal extent which makes it more suitable for detection by the MLS instrument. This is supported by that fact that in contrast to Case 1, both the ascending and descending profiles show enhancements of tropospheric air in the lower stratosphere. 340 The trajectories for the two cases show an increase in water vapor before the air parcel arrived at the measurement site. The increase in water vapor is accompanied by an increase in PV and high CAPE values. While in Case 1 the steadily decreasing ozone values along the trajectory seem to be unrelated to the changes of the other trace gases, in Case 2, an increase in ozone mixing ratios by 150 ppbv occurs at the same time as the increase of the PV values. In contrast to Case 1, where the peak in PV initially decreases shortly before reaching the measurement site and returns to background values a day later, in Case 2 the PV 345 values keep increasing but never reach the high values of Case 1. In both cases, the water vapor mixing ratios remain enhanced





after the overshooting convection in the model, shortly before reaching the measurement site. However, Case 2 shows lower values around 10 ppmv in comparison to 15 ppmv for Case 1. With these values the ERA5 water vapor value is greater than the measured value in Case 1, but is slightly below the values measured in Case 2.

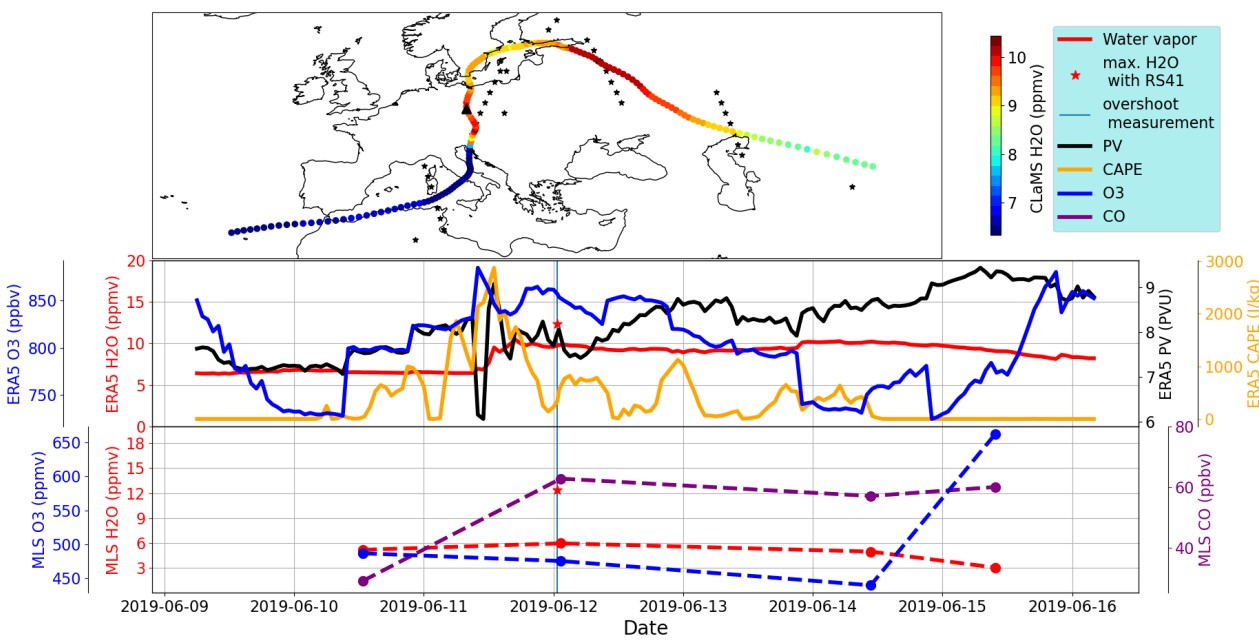

**Figure 10.** Same as in Figure 9, but for Case 2.The time of the measurement and the observed maximum water vapor mixing ratio from CFH and RS41 within the pressure levels of 139 hPa and 155 hPa are shown with a vertical blue line and with red symbols, respectively.

## 3.6 Overshooting events in satellite data

The satellite measurements of brightness temperature (BT) from geostationary Meteosat-10 rapid scan data support the above-indicated tropospheric origin of the measured water vapor enhancement in the lower stratosphere. Figures 11a-d show Meteosat-10 BT data for two chosen times in Case 1. Panel a in Figure 11 shows the data at 05:29 UTC on 10 June 2019. A cloud structure reaching a BT as low as 205 K surrounds the air mass along the trajectory at this time. For Case 1, a CPT of 208 K was measured and confirmed by the surrounding cloud-top BT between 210 K and 216 K. It is therefore most likely, that areas with a

BT below 205 K resemble areas of overshooting tops. These areas are circled in pink in Figure 11. In addition to the trajectories discussed in section 3.5, further trajectories were calculated starting at the same location but, at lower pressure levels, as both balloon profiles not only exhibited a main peak at a pressure level of 149/144 hPa but also covered an underlying water vapor enhancement at 165/155 hPa respectively, for Cases 1 and 2 (Figures 3 and 4). Trajectories initialized at 149, 155 and 165 hPa





and at 144, 153 and 155 hPa respectively for Case 1 and Case 2 were calculated and added to the satellite image. The air mass
on the trajectory starting at 149 hPa is located closest (only 50 km north-east) to the coldest and therefore, highest point of
the convective cloud, as can be seen in detail in panel a of Figure 11. Considering the slight uncertainties in the trajectory
calculation and in the meteorological fields, this point in time is most likely responsible for the water vapor enhancement
detected later. However, the satellite images display the coinciding of the air mass and the convective event 4 hours earlier
than compared to the ERA5 data and therefore, further south-west. Later in the day the air mass location coincides with an-
other overshooting cloud around 21:09 UTC (see Figures 11b). Multiple areas, exceed the tropopause height in the convective
clouds but none of the air masses on the trajectories seems to be very close to these areas. The trajectories for Case 2 pass
near convective events as well, however in greater distance (Figure 12a and b). For Case 2, Figure 4b shows a temperature
of 214 K at the tropopause height. Trajectories initialized at a pressure level of 145 hPa, 150 hPa and 155 hPa can be seen in
Figure 12 while encountering cloud-top height with temperatures 6 K below the tropopause temperature. Similar to Case 1,
in Case 2 a convective storm develops additionally over eastern Germany with overshooting tops. However, the air masses
along the trajectories do not encounter this convective cloud (see panel Figure 12a and b). Thus, it is very likely that the water
vapor enhancement resulted from the overshooting event that occurred over Austria on 11 June 2019 at around 14:24 UTC or
15:49 UTC.

## 4   Discussion

The measurements presented here show a strong enhancement in water vapor above the tropopause for two consecutive days.
Both cases originate from convective overshooting into the lower stratosphere and water vapor was injected several hours
before the balloon launch.

The water vapor mixing ratio enhancement measured in Case 1 is located 40 K above the thermal tropopause, when using
potential temperature as a vertical coordinate. This is comparable to a study by Smith et al. (2017) where water vapor mixing
ratio enhancements were measured during multiple airborne missions above the North-American continent. Smith et al. (2017)
use 370 K as a typical tropopause altitude and discuss water vapor enhancements at a level between 400 and 410 K with values
up to 6 ppmv above the background values. Similarly, the water vapor values measured in Case 2 are in the same order of
magnitude. The maximum of the peak is approximately 40 K above the tropopause potential temperature and reaches 7.5 ppmv
above the background value. The same order of magnitude was observed during the $SEAC^4RS$ aircraft measurement campaign
with elevated water vapor mixing ratios of up to 10.6 ppmv in the lowermost stratosphere at $\approx$ 100 hPa (Robrecht et al., 2019).

Parallel to the local injection of water vapor a peak in ozone was detected for Case 1. This peak results from a horizontal
transport of stratospheric air masses with a strong stratospheric signature from west to east. An edge of a filament from a front
with high ozone values is stretched over the measurement location. Figure 6 shows that according to ERA5 data, where the
measurement was at the edge of an air mass which moves ahead of a front with higher ozone mixing ratios at the 145 hPa level.
This explains the lower ozone values at the same pressure/potential temperature level in the descending profile further north
and is supported by the sparse data from MLS, which shows higher ozone mixing ratio values westward of the measurement



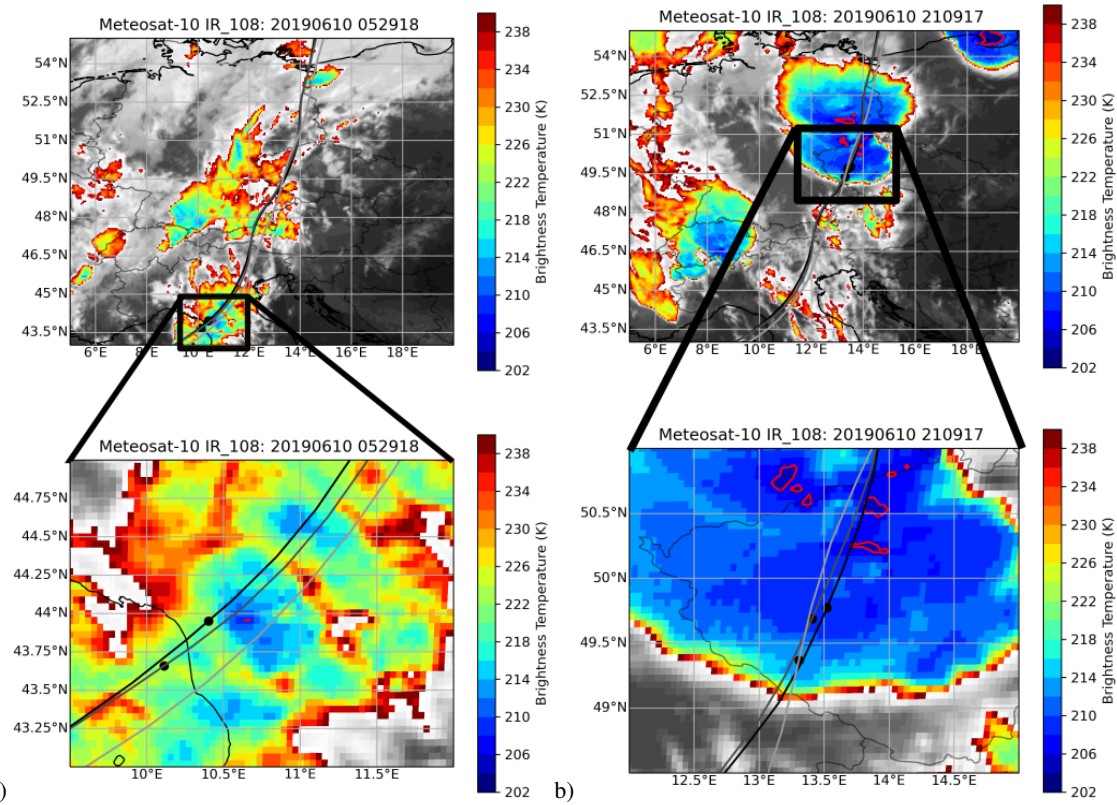

**Figure 11.** Brightness temperatures from geostationary METEOSAT10 satellite using the IR 10.8 $\mu$m channel along trajectories for Case 1 during two different times (panel a at 05:59 UTC and panel b at 21:09 UTC on 10 June 2019). Air mass trajectories initiated at different pressure levels (149 hPa, 155 hPa and 165 hPa) are shown with gray to black lines. Air masses with BT < 205 K are depicted with pink contours.

site and lower values of about 200 ppbv easterly of the measurement site (see Figure 9). The ERA5 ozone values along the calculated trajectory of the measured air mass further support this assessment. The moistening of the ozone-rich air mas lead to an unusual feature in the tracer-tracer correlation, as shown in Figure 5b.

Case 1 not only shows a strong enhancement of water vapor mixing ratios in the ascending profile of the balloon-borne measurement, but further expected indications of a tropospheric air injection were also recorded. A sharp decrease in ozone mixing ratios occurs at the same potential temperature level as the rise in water vapor. The drop in temperature is equally sharp and aligned with the change in water vapor and ozone, albeit less prominent. The elevated water vapor, decrease of ozone mixing ratios, and lower temperatures all indicate the tropospheric origin of the measured air mass between the potential

temperature levels of 365 K and 375 K. The air mass is clearly different from the air masses above and below to a degree that the profile of the water vapor peak appears to be square-shaped (see Figure 3b). The fresh in-mixing and the tropospheric origin of the air masses is also underlined by the small spatial extent of the enhancement. This is derived from two observations: Firstly,

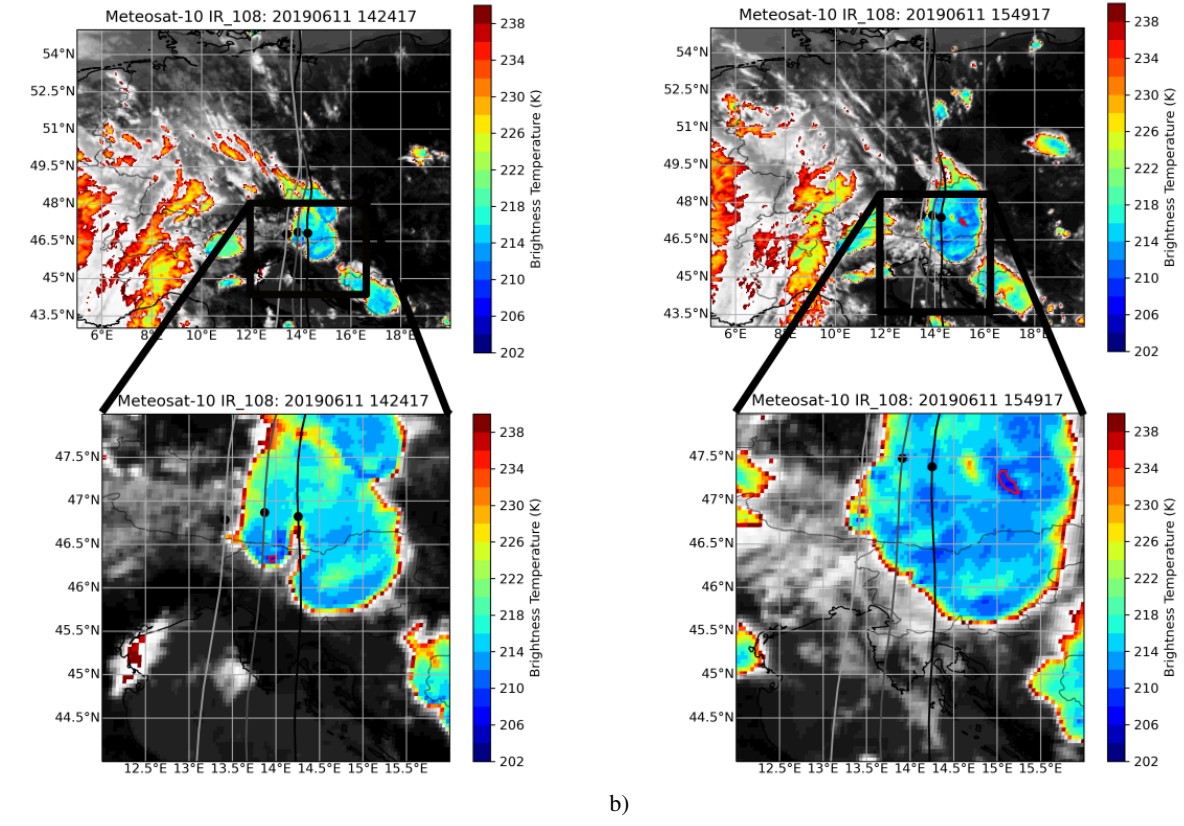

**Figure 12.** Same as Figure 11 but for Case 2. Air mass trajectories initiated at different pressure levels (144 hPa, 153 hPa and 155 hPa) are shown with gray to black lines. Air masses with BT < 209 K are denoted with pink contours. Panel a shows the satellite image at 14:24 UTC and panels b at 15:49 UTC on 11 June 2019

the balloon measurement does not show any enhancement of water vapor in the descending profile and secondly, no clear trace of the event can be found in the MLS measurements, due to the low vertical and horizontal (cross-track) resolution at the limb tangent point of 1.5 km and 3 km, respectively. The tropospheric source of the water vapor injection for Case 1 is supported by satellite BT measurements of overshooting tops as well as ECMWF ERA5 data displaying a local disturbance in PV and dTheta along the trajectory of the discussed air mass. The dTheta anomaly is in good agreement with the local enhanced water vapor mixing ratio in ERA5 (Figure 7d - f). The BT satellite data suggests this coinciding of the measured air mass and the convective overshooting event 4 h earlier at 05:29:18 UTC further south-west but otherwise both data sets exhibit the same behavior of a convective storm moving northwards with high-reaching cloud tops and BTs reaching as low as 205 K. Until this convective event dissolves after 22:00 UTC, the measured air parcel remains close to its center. The air mass along the trajectory starting at 149 hPa encounters a second, stronger and more spatially distributed convective event over the north-east of Munich (south east Germany) in the evening at 18:19 UTC and 19:39 UTC where it also passes close to a cloud-top height reaching 202 K BT





(see Figure A2). The air masses continue to remain in this growing convective cloud, which eventually covers the entirety of
eastern Germany, until the measurement site is reached. As described in Smith et al. (2017), Qu et al. (2020), and Dauhut et al.
(2018), the injection of tropospheric air masses is followed by the mixing of the entrained air masses with the surrounding
stratospheric air, which has a lower water vapor mixing ratio and a higher potential temperature. The hydro-meteors from the
injection sublimate and are mixed into the stratospheric air mass on very small time-scales under these strongly sub-saturated
conditions. It is therefore consistent that the additional COBALD measurement (not shown) did not detect any cloud particles
in the measured profile. It is very likely that the air mass descended slightly due to the decreased potential temperature after
the mixing of tropospheric and stratospheric air and therefore reached neutral buoyancy at a lower level as was later found in
the balloon profiles. This process would not be evident in the calculated trajectories and thus slightly increases the inaccuracy
of the presented trajectories. However, the descent of the air mass due to the adjustment of potential temperature is expected to
be rather low due to the very humid conditions in the overshooting top and the existence of sublimated hydro-meteors in the
entrained air masses which result in a relatively low amount of air that is ultimately irreversibly mixed within the LS.

Case 2 shows a similar signature but differs in several aspects. First, the balloon profile measurements in Figures 4a and
b show that the water vapor enhancement is stronger and is located at a lower pressure level (although the level of potential
temperature remains almost the same). In contrast to Case 1, only radiosonde measurements are available for Case 2 and thus
only water vapor data measured by the radiosondes can be compared for both cases. While in Case 1 the peak value is 7.0 ppmv,
in Case 2 the maximum peak value reaches up to 12.1 ppmv. In addition, balloon measurements for Case 2 potentially indicate
not only a stronger but also a spatially larger event, as the descending profile reveals a peak in water vapor that still reaches
more than 7.5 ppmv but with a vertically wider spread. This indication is supported by the sparse MLS measurements. For Case
2, a slight increase of MLS water vapor mixing ratios ($\approx$1 ppmv) is visible after the balloon observation compared to the MLS
data point that was taken before the suggested convection event (see Figure 10). The development of the water vapor peak does
not form a square shape when using potential temperature as a vertical coordinate, instead forming a sharp tip after a steep
water vapor increase. Shortly below this tip a drop in temperature can be seen (Figure 4b), with a similar decrease (by about
2 K) as in Case 1. A further difference between the cases is the tropopause. While Case 1 shows a very sharp tropopause, Case
2 has a rather flat tropopause with an inversion layer at 125 hPa. In the temperature profile obtained 2 h prior to the profile with
enhanced water vapor (displayed in Figures 4a and b) a second tropopause is detected. Throughout the day of Case 2 (launched
at 13:11 UTC and 22:00 UTC on the 11 June 2019) two profiles with double tropopause were measured, which indicates a
less stable atmospheric profile and possibly supports the findings of Solomon et al. (2016) that found that the overshooting
convection is more likely in cases of a present double tropopause.

ERA5 data also show a difference between the two cases. Case 2 shows a higher and wider spread of CAPE values before
the overshooting event throughout the day compared to Case 1. The PV anomaly discussed in section 3.4 shows a cluster of
individual anomalies around the calculated trajectory instead of a single event as in Case 1. According to the satellite data the
air mass measured in Case 2 encounters a convective event once in the afternoon over Austria (see section 3.6), but does not
encounter a second event later in contrast to Case 1. While ERA5 shows the overshooting event for Case 1 4 hours before it

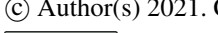



was observed with satellites slightly more south-east, it shows the convective event for Case 2, 4 hours later than indicated by ERA5.

Overall, the ERA5 data indicates that the measured air parcels of both cases were moistened by the occurrence of an overshooting convective storm, which caused a local not conservative PV anomaly to appear at the level of the measured air parcel. This PV signal was most likely caused by the upward displacement and narrowing of the isentropes in combination with diabatic processes and related small-scale mixing (Qu et al., 2020). The overshooting convection which moistened the measured air parcel in the lower stratosphere occurred in the north of Italy several hours before the convective event arrived

at the measurement site as implied by the ERA5 and satellite data. However, the moistened air parcels took a similar pathway in the lower stratosphere as the convection in the troposphere before both arrived at the measurement site in eastern Germany. It is not clear from the data whether the air parcel was moistened once during the first appearance of the convective storm in the northern part of Italy or was moistened multiple times along the trajectory as indicated by the satellite images over Bavaria south of the measurements site. In addition, the question as to when exactly water vapor was injected into the lower

stratosphere remains unanswered. However, it is evident that the water vapor was injected into the lower stratosphere by convective overshooting and is not caused by for example horizontal transport and the in-mixing of tropical air masses at the sub-tropical jet.

## 5   Conclusions

Overshooting convective events are known to inject water vapor into the lower stratosphere. However, their quantitative impact

on the variability of water vapor mixing ratios in the mid-latitudes requires further investigation. A number of case studies of overshooting tops and their vertical transport of water vapor were performed above the North American continent. However, in-situ measurements over Europe are still sparse. In this study, two cases of water vapor injection into the lower stratosphere over the German-Czech border are presented. The balloon-borne in-situ measurements show water vapor enhancements related to the background value of 5 ppmv by 3.65 ppmv for Case 1 and 7.1 ppmv for Case 2. Both cases show clear evidence for

overshooting events and have a comparable scale to overshooting events measured in previous studies over the North-American continent. The findings of the in-situ balloon borne measurements are supported by ECMWF ERA5 data as well as by satellite data. We emphasize here that ERA5 data includes the overshooting convection and moistening of the LS in both cases in contrast to ERA-Interim reanalysis data. The location and timing of the observation was not precisely matched by ERA5, but was nevertheless relatively close to the event observed by the satellite data. It is shown that the measured enhancements of

water vapor at a pressure level of 149 hPa and 144 hPa respectively, for Cases 1 and 2 were injected into the lower stratosphere several hours before the measurement took place and were horizontally transported to the measurement site. Stratospheric moistening through overshooting convection over the North-American continent has already been discussed, but it is also measured over Europe in our study. The strength of the measured water vapor enhancement measured shows that the role of overshooting convection over Europe and in mid-latitudes in general as a contributor to the lower stratospheric water vapor

budget might be underestimated due to the sparse in-situ data. As it is expected that the frequency and strength of extreme



convective events will increase with advancing global climate change, it is crucial to understand and quantify the impact of these events thoroughly. MLS satellite measurements are not always suitable for detecting these small-scale water vapor enhancements, as shown especially in one of the two cases. Thus, studies estimating the relevance of overshooting convection on the extra-tropical water vapor distribution using satellite data might underestimate their effect in general and not only over 485 Europe alone. Due to the low resolution of satellite data, in-situ measurements are therefore required to understand the impact of such small-scale events like overshooting convection.

*Data availability.* Balloon-borne data: https://ddp.tereno.net/ddp/dispatch?searchparams=freetext-Balloonmetadata.detail.view.id=ae5659be-5b53-43ea-82a9-61dfca3263edmetadata.detail.view.origin=EIFEL.RURmetadata.detail.view.type=org.fzj.ibg.catalog.shared.ISO19115Bean

MLS data: https://earthdata.nasa.gov/earth-observation-data/near-real-time/download-nrt-data/mls-nrt

ECMWF ERA5 data: https://www.ecmwf.int/en/forecasts/datasets/reanalysis-datasets/era5

## Appendix A:  Instruments used on the balloon payload

### A1  Electrochemical concntration cell (ECC)

Electrochemical concentration cells (ECCs) are light weight in-situ ozone sondes that have been used for multiple decades on weather balloons to investigate ozone mixing ratio profiles and to monitor long-term ozone trends, for example in the southern 495 hemisphere within the SHADOZ network (Southern Hemisphere ADditional OZonesondes) (Thompson et al., 2019). The ambient air is pumped through a Teflon tube into the reaction cell at a speed of about 29 s/100 ml at ground conditions. In the reaction cell of the device a chemical reaction of the ambient ozone with potassium iodide produces two electrons for each ozone molecule. The resulting current is thus proportional to the partial pressure of ozone in the sampled air. Therefore, the ECC does not need a calibration procedure prior to the balloon launch. Komhyr et al. (1995a) describe the ECC in detail. In 500 order to gain high-quality data, the exact composition of the potassium iodide solution is crucial. Johnson et al. (2002) present an extended study of different solution variations and their effect on the background current. In this study we use a solution composition of 1/10 and a full buffer suggested by Johnson et al. (2002) for the most accurate ozone data in contrast to the long-term measurement series that consistently uses one solution combination suggested by (Komhyr et al., 1995a,b; Smit et al., 2007).

### 505  A2  Cryogenic frost point hygrometer (CFH)

The cryogenic frostpoint hygrometer (CFH) is a balloon-borne instrument based on the cold mirror principle, which regulates the temperature of a mirror to the frost point temperature of the ambient air (Vömel et al., 2007) and therefore it does not require calibration. The CFH is considered to be one of the few instruments capable of measuring stratospheric water vapor (Nash et al., 2010). The mirror is constantly cooled with a cooling agent, R-23 (triflourmethane), and a feedback loop regulates 510 the temperature of the mirror to be constantly coated with a thin layer of ice . Prior to the flight, the cooling agent is precooled





to liquid state at -86 °C and poured into the instrument. With decreasing ambient pressure during the flight, the temperature of the liquid decreases to -120 °C at tropopause level. When the mirror is covered with a thin layer of ice and is in steady state, the mirror temperature is equal to the frost point temperature of the measured air mass. The thickness of the ice layer is controlled by the reflectivity of the mirror, which is measured by a light beam and a detector. As soon as the ice layer on the

mirror becomes thicker, a smaller proportion of the light beam is reflected by the mirror. The detector then sends a signal to a heater to regulate the temperature of the mirror. This regulating cycle is constantly maintained, leading to a thermodynamical equilibrium between ice layer and ambient air and resulting in a relatively low uncertainty of the instrument. The uncertainty of the instrument is defined by the stability of the feedback controller and is defined as 0.5 °C, which leads to a relative unvertainty below 4% in the troposphere and below 10% in the stratosphere (Vömel and Diaz, 2010; Vömel et al., 2016). Klanner et al.

(2021) used the CFH in comparison with the water vapor lidar system and found a very good agreement within their respective uncertainties throughout the entire atmospheric profile. A potential failure of the instrument can be caused by liquid droplets on the detector, the mirror, or the light source. It is therefore recommended that the CFH is not launched during rain.

## A3 Vaisala Radiosonde RS41

The Vaisala Radiosonde RS41 was introduced in 2014 and completely replaced the RS91 precursor model in 2017. The

temperature sensor is based on resistive platinum technology. The manufacturer states that there is a combined uncertainty of 0.3 K below 16 km and of 0.4 K above. The response time of the temperature sensor is < 1 s and thus does not need to be considered in the following. The temperature range is given as -95 °C to 60 °C and the resolution is 0.01 °C (Jauhiainen et al., 2014; Vaisala, 2020). The humidity sensor is a thin-film capacitor. The combined uncertainty for the humidity sensor is given as 3% and the resolution is given as 0.1% relative humidity. Similar to the temperature sensor, the response time of the humidity

sensor is < 0.3 s at 20 °C and < 10 s at -40 °C. The pressure sensor is a silicon capacitor and is defined for a pressure range between surface pressure and 3 hPa while the resolution is given as 0.01 hPa.

Dirksen et al. (2020) find in experimental work that the humidity sensor of the RS41 has an uncertainty of < 1.5% and a temperature uncertainty of < 0.2%. Marwati et al. (2008) validate the uncertainty of the RS41 temperature and humidity data. No systematic drifts due to storage were found either in temperature or in humidity measurements. However, an increase of

temperature uncertainty was found from 0.13 °C in the troposphere up to 0.3 °C at 30 km. An uncertainty below 1% relative humidity was found in the stratosphere.

*Author contributions.* CR, DK and AW performed the balloon-borne measurements and wrote the draft. CR and DK performed the data analysis from the balloon-borne data, the MLS data and ECMWF ERA5 data. CR performed the Meteosat-10 data analysis. Data interpretation of the ECMWF ERA5 was performed by JG, RM, PK, CR and DK. All authors have contributed with scientific discussions to data

interpretation, read and agreed to the published version of the manuscript.





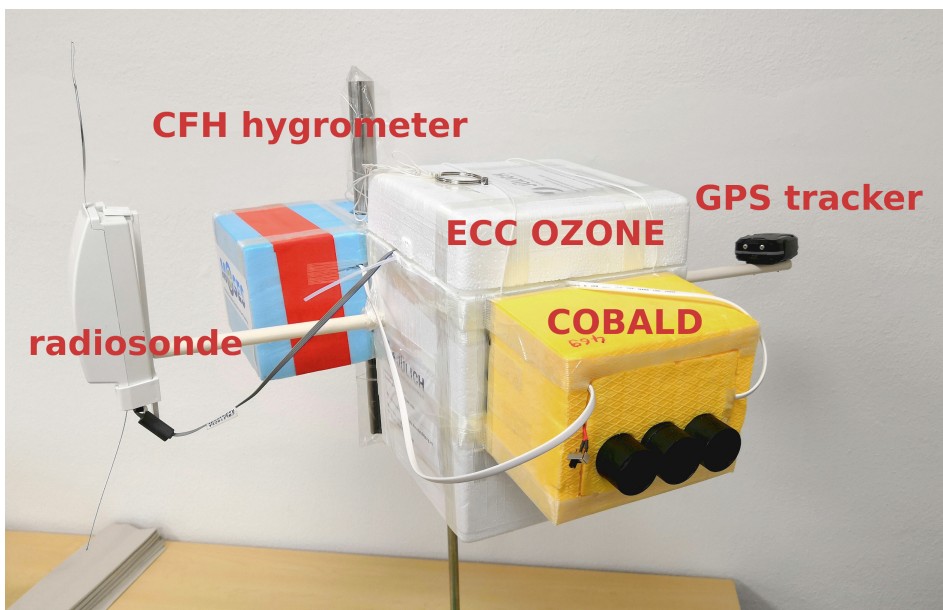

**Figure A1.** Combined full balloon payload. The ECC is located in the white styrofoam box, the COBALD is in the yellow box and the CFH is embedded in the light blue box. A white wooden stick goes through the ozone box and has the radiosonde attached on one side, while an independent GPS tracker is attached on the other to locate the payload after the landing. In total the payload has a mass of 2.6 kg. The payload is attached to an unwinder, which ensures that the balloon launch takes place smoothly despite the 60 m string between the balloon and payload. The unwinder is connected to a parachute to reduce the falling speed of the payload after the balloon bursts and assures a safe landing. The parachute is connected to the 1500g balloon.

*Competing interests.* The authors declare that they have no conflict of interest.

*Acknowledgements.* The balloon activities were funded by the Helmholtz Association within the framework of MOSES (Modular Observation Solutions for Earth Systems). We would like to thank NASA and the MLS team for providing the water vapor data from the Microwave Limb Sounder (MLS) on the Aura satellite. We would also like to thank EUMETSAT for the Metop10 brightness temperature data. In addition, we are grateful to the ECMWF for their meteorological reanalysis data support. A special thanks goes to the Karlsruhe KIT-cube Team who always supported us during the MOSES measurement campaign.






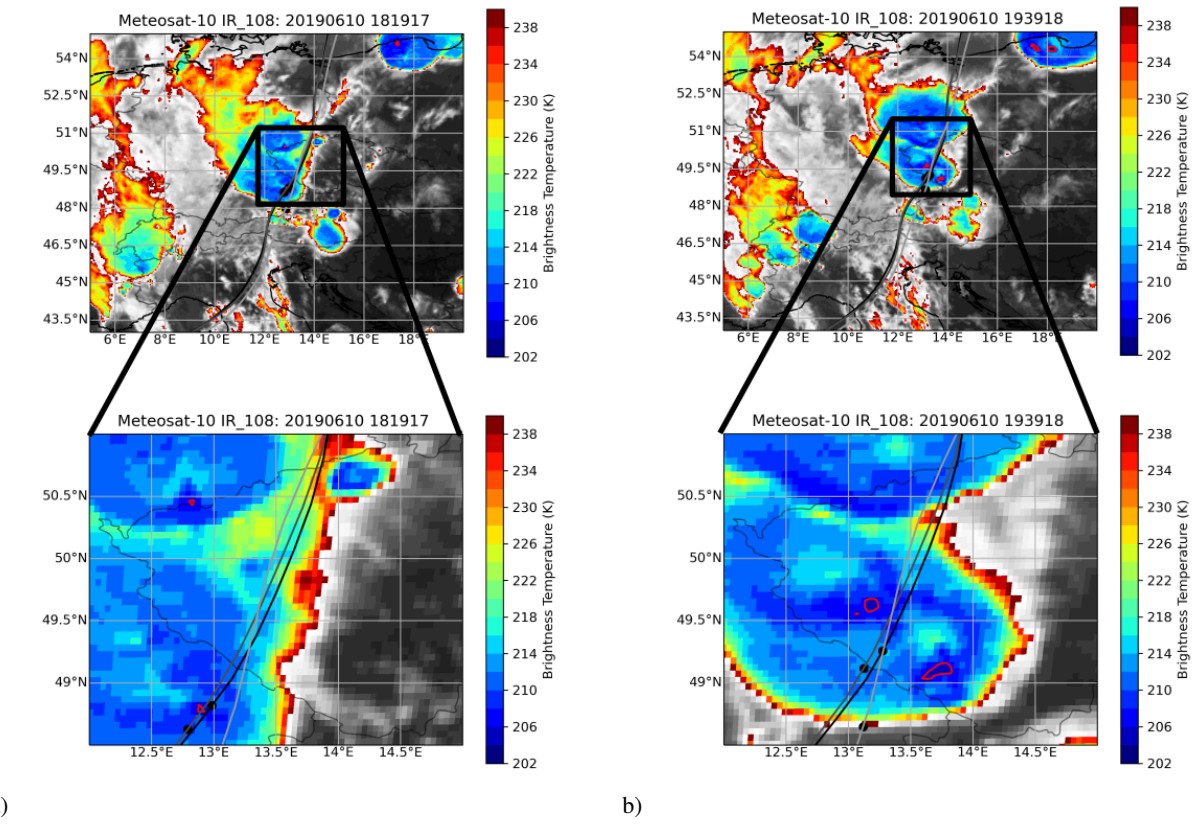

a)                                                                    b)

**Figure A2.** Same as Figure 11 but for a different point in time. Air mass trajectories initiated at different pressure levels (144 hPa, 153 hPa and 155 hPa) are shown with gray to black lines. Air masses with BT < 209 K are marked with pink contours. Panel a and c show the satellite image at 18:19 UTC and panel b and d at 19:39 UTC on 10 June 2019

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
