# Peer review of "A case study on the impact of severe convective storms on the water vapor mixing ratio in the lower mid-latitude stratosphere observed in 2019 over Europe"

_Atmospheric Chemistry and Physics, 2021_

## Referee Comment (RC2)

Khordakova et al., 2021, describe two instances of water vapor enhancements observed in the lower stratosphere over Europe. The study is motivated by measurements obtained with balloon-borne instrumentation. Satellite observations of atmospheric trace species as well as convective activity, and air parcel trajectory calculations combined with output from global reanalyses are utilized to understand the source of these individual hydration events within the broader dynamical and chemical context. In both cases, the hydration events are tied to recent convection near the measurement site.

Main comments/concerns/questions:

- Regarding the relative levels of the water vapor and ozone peaks: Are sampling time constants for each instrument accounted for? If the response time for water/CFH is slower than that for the ECC cell, I can imagine a case where their peak locations might be offset?

- The structure of the temperature profiles, and the mixing ratio profiles when converted to potential temperature are peculiar (square-shaped and peaked in Cases 1 & 2, respectively), is it possible there is a measurement issue - either vertical resolution, or something else?

- I am not convinced by the argument given in Lines 237+ regarding the profile of Case 1 being within the "usual range of observations." This may be true, but with only 8 profiles in total, it's difficult to define a "usual range." It might be possible to obtain better statistics on the distribution of ozone mixing ratios at this level using MLS?

- How narrow (vertically) are the modeled ozone enhancements in CLaMS and the ERA5 output? Is it consistent with the exceptionally narrow (~0.6 km) enhancement in ozone observed on the ascent profile?

- Lines 244-247: The brief mention of a model run with CLaMS "using different ECMWF data sets as input" is hard to evaluate and understand, particularly as the results are not shown.
  What is shown is a plot of output from ERA5, which, as the authors state, shows evidence for convective influence in multiple variables.
  More regarding the ERA5 ozone:
    o Why does the map of ERA5 ozone (Fig. 6) with peak values at 145 hPa of ~640 ppbv to 560 ppbv, differ so much from the ERA5 trajectory values in Fig. 9, which are >800 ppbv for the time period of interest?
    o Is it reasonable to assume the assimilated ozone is impacted by convection?
    o Though ozone declines over the trajectory (Fig. 9), the declines do not appear to be closely associated with the locations/timing of convection. This is also noted by the authors in Lines 320-321.

- While I think this paper does a good job of utilizing and even showcasing the value of the new ERA5 product, it's somewhat concerning that the ERA5 ozone values are so elevated relative to measurements – they're consistently ~80% higher!

Additional comments:

- I find the practice of referring to reanalysis output as "data" misleading, as it is not a primary observation/measurement.
- Statements such as those in Lines 507-508 and in Lines 498-499 about instruments not requiring calibration are misleading. It is best to leave such statements out of a paper like this where there is not room to fully understand their meaning and context. A reference to papers describing the development, operation, calibration and validation of the ECC ozonesondes and CFH and other chilled mirror hygrometers is sufficient.
    - Ozonesondes do not need a calibration with ozone standards before they are launched, however, they do need to go through careful preparation one to two weeks prior to launch. These steps include high ozone conditioning the pump/tubing, measuring the background cell current, accurately measuring the teflon pump flow rate, using the correct solution and understanding the pump's efficiency over the vertical profile, among many other things.
    - Similarly, while CFH does not need a calibration with water vapor standards prior to launch, it does require a thermistor calibration among other operational assessments to ensure measurement accuracy.
- Similarly, I believe what is meant in Lines 508-509 is that there are few *balloon-borne in situ instruments* capable of measuring stratospheric water vapor. There are balloon-borne remote sensing instruments, as well as satellite instrumentations, e.g., MLS. Furthermore, there are multiple in situ instruments employing different techniques for measuring stratospheric water vapor aboard aircraft.

Minor comments/edits:

Abstract

- Line 14: …values *measured* by MLS in the LS are lower than the in situ observations… (MLS observations are measurements not estimations)
- Line 14: …ERA5 overestimates water vapor mixing ratios… (fine to say that reanalysis overestimates a value, given that it is not a primary measurement/observation)
- Line 16+: This is in good agreement with the reanalysis which shows a strong change in the structure of isotherms and a sudden and short-lived increase in potential vorticity at the altitude of the trajectory. Similarly, satellite data show low cloud top brightness temperatures during the overshooting event. (clearer delineation between the satellite observations and the reanalysis derived output)

Introduction

- Line 27: …entry mixing *ratio*…
- Lines 30-34: It may be worth mentioning that there is some debate about the net magnitude of SWV feedback, with some studies suggesting that the impact on climate sensitivity may be significantly lower if changes in SWV are evaluated within a coupled system, e.g., Huang et al., 2020.
- Lines 43-44: The observational analysis of Smith et al., 2017, focused on a significant *localized* effect. The model analyses of Dessler et al., are looking at global impacts. (By the way, Dessler et al., 2013a is the same as Dessler et al., 2013b.)
- Line 56: …North American and Asian Monsoon *regions*.
- Line 61: …the data *is/are* compared to output of the ECMWF ERA5 reanalysis, while…

Section 2 Data and Methods

- Line 76: (Consistency in tenses.) …before the convective cell *reached* the…
- Line 81 & later, Lines 98-99: The actual uncertainty of the Vaisala humidity sensor is found to be far greater than 3% through comparison with CFH in the altitude range of interest. This is evident both in Figure 2, where differences can be ~100% at altitudes <20 km (log/log plot), and in Figure 3, where differences in the quiescent stratosphere are on the order of 30-50%. That said, for the purposes of this analysis, the absolute accuracy of the Vaisala sensor is less important than its sensitivity to detecting abrupt changes with a magnitude far greater than its measurement uncertainty.
- Line 84: …ozone mixing ratios *with* an uncertainty…
- Line 87: …uncertainty of the CFH instrument *is* given as…
- Line 103: …stratosphere *reaching* altitudes of up to…
- Line 106: …but all other sounding data *are* complete.
- Line 108: What does "a local instrument" mean? MLS is a remote sensing instrument.
- Line 111: *Temperature and pressure are retrieved*…

Section 3.2 Water Vapor Injection – Balloon

- Line 182: (Awkward.) *A background value of ≈5 ppmv agrees well with*…
- Line 185-186: (Given the ~±1 ppmv differences between the Vaisala and CFH in the quiescent stratosphere, i.e., above the hydration layer, I am not convinced the response time is the sole cause of the difference in the plume.) You might say: *The lagging response time of the RS41 may explain most of the difference between the CFH and the radiosonde observations.*
- Line 191: (I think you could start a new paragraph for the discussion of the ozonesonde and temperature profile results.) Consider: *A striking peak in the ozone profile is evident at a similar level as the peak in water vapor.*
- Line 204: (I think you could start a new paragraph for the discussion of Case 2.) Also: …different background atmosphere *than* Case 1.
- Line 205: …as *is depicted* in Figure 4a…

Section 3.3 Source of Ozone Peak

- Line 227: (Delete "that") …*shows a steep* decrease…
- Line 240: (Delete "but") …profile at this *altitude, it* remains…
- Lines 241+: (The term "measurement data" is vague.) Consider: *Here, the data from Case 1 (red dots) with the high ozone and water vapor diverge…*

Section 3.4 Comparison with ER5

- Line 290: Consider: …diabatic heating, *which cause* an increase…
- Line 301: Consider: *Similar to* rather than *Similarly to*

Section 3.5 Origin & Evolution Along CLaMS

- Lines 304+: For simplicity, I think it would be best to refer to the trajectory output as ERA5 consistently rather than switching back and forth between ECMWF, ECMWF ERA5, etc.
- I found this confusing and am wondering how to reconcile the "multiple MLS data points" from Line 311 with "Only 7 measurement points" from Line 323? I'm also unclear how

the dots in Fig. 9 panel C correspond to the overpasses in panel A? It looks like there are 9 distinct overpasses that cross the ERA5 trajectory, but only 7 points in panel C?

Section 3.6 Overshooting in Satellite Data

- Line 365: (No comma needed after "Multiple areas")
- Line 367: Consider: "…*however, at a greater* distance…"

Section 4 Discussion

- Line 375: Consider: "...*on* two consecutive days…"
- Line 376: Consider: "…overshooting *that injected water vapor into the lower stratosphere* several hours…"
- Line 386: Consider: "*The local injection of water vapor was detected within a larger scale peak in ozone for Case 1.*"
- Line 388: Consider: "*A map of ERA5 ozone at 145 hPa in Figure 6 shows that the balloon measurement was at the edge of a front with higher ozone mixing ratios. This explains the lower ozone values at the same pressure/potential temperature level in the descending profile, which was located further north. This is also supported by the sparse data from MLS, which show higher ozone…*"
- Line 393: "…ozone-rich *air mass*…"
- Line 409: Consider: "…further south-west*, but otherwise both the reanalysis and the observational data show a* convective storm moving northwards…"
- Line 434: "…before the suggested *convective* event…"
- Lines 447+: ("it" is vague, I wasn't sure if "it" referred to ERA5 or the satellite data, and whether you meant to write satellite data at the end of the sentence?) Consider: "*While ERA5 shows a likely overshooting event for Case 1 four hours before and slightly north-west of the event observed by satellites, the convective event in ERA5 for Case 2 is four hours later than the satellite observations.*"
- Line 461: Delete "for example" or consider: "…and is not caused by another mechanism, for example, the horizontal transport and…"

Conclusions:

- Line 468: Consider: "…water vapor enhancements *in excess of* the background…"
- Line 478: ("measured" is used twice.) Consider: "…measured water vapor *enhancement shows*…"

A2:

- Line 518: ("uncertainty" is misspelled.)

Figures

- Figure 1: …after a deep convective event has passed the *measurement* site. On the right hand *side* the…
- Figure 5: Why do the lines corresponding to the ascent/descent profile exceed the gray area, which "marks the measured range of ozone mixing ratios?" Were these two profiles excluded from the definition of the mean and the range?

---

## Author Comment (AC1)

We would like to thank the referees for their time and investment to go through the paper and give valuable constructive and detailed comments. We created a revised version of the article for re-evaluation including all suggestions, improved unclear formulations and corrected Figure labels. We finally thank the reviewers in advance for their renewed efforts.

We have reanalysed the ozone data for the measurement results of Case 1 according to the new publication of Vömel et al. 2020. This confirmed the main message of the paper that tropospheric air masses with lower ozone and higher water vapor values were mixed into the stratopheric air by gravity wave breaking behind an overshooting top. In the updated Figures, the drop in ozone and peak in water vapor coincide almost perfectly. Further a unit conversion error in the analysis of ozone data along the trajectory from ERA5 data was corrected.

10 In the following we address all the comments by the two referees.

**1 RC 1**

15

25

30

- L194-196: To make this phrase easier to understand, I suggest to mention first the enhanced ozone and point to the subsequent section 3.3. Other wise, the description here appears to me difficult to understand. I see directly the high ozone content rather than the diluted ozone due to the mixing with tropospheric air. The question also rises here why the ozone content is high

 $\rightarrow$  Corrected, Changed to: "A striking peak in the ozone profile is evident at a similar level as the peak in water vapor. With a lower edge at 162 hPa/359 K and an upper edge at 145 hPa/373 K the ozone peak starts at a lower level compared to the water vapor enhancement, but is limited by the same upper edge. This ozone peak is not associated with the overshooting event and the cause is discussed in Section 3.3."

20 - L201-202: I'm wondering what is the cause of the zigzag form of the T profiles between 160 and 140 hPa. Is this due to the instrument artifacts or any other reasons?

 $\rightarrow$  We think it is not an instrument artifact, as temperature measurements on radiosondes are quite accurate even at low temperatures (WMO Intercomparison of High Quality Radiosonde Systems, 2011). The drop in temperature is suggested to be caused by in-mixing of tropospheric air from the overshooting top which is much colder due to adiabatic ascent despite contributions from latent heat release.

- L202-203: I have doubt about the use of overshooting top in this statement, The overshooting tops are usually of small dimension (several km in D). The ice water content is always very height inside and the temperature are usually much lower than the ambient lower stratospheric air (much larger than 2K). In addition, within the overshooting top it is usually very dry due to the low temperature. I guess what you talk about here is either the mixing area near the overshooting tops due to the breaking of gravity waves (jumping cirrus, ice plume and eventually plume with higher humidity, etc.). I do see in the simulation that these areas can be a little cooler that the ambient temperature due to the mixing with lower troposphere air and the sublimation of mixed ice just after the convection. Same for an other statement in L424

- → We agree that the wording is misleading and changed the text to '[...]result of mixing with the strongly idiomatically
   cooled tropospheric air within the overshooting top and the warmer stratospheric air masses in the surrounding. In addition, evaporation/sublimation of cloud particles in this warmer and dryer mixing area around the overshooting top can also lead to further cooling.' and in L424: '[...] the in-mixing of tropospheric air masses caused by gravity wave breaking closely behind an overshooting top into the surrounding stratospheric air, which has a lower water vapor mixing ratio and a higher potential temperature. The hydro-meteors from the injection sublimate and are mixed into the stratospheric air mass on very small time-scales under these strongly sub-saturated conditions. It is therefore consistent that the additional COBALD measurement (not shown) did not detect any cloud particles in the measured profile. The sublimating hydrometeors additionally cool the air mass.'
  - Figure 34 is there any ice observed in these two cases? If there is any in the lower stratosphere, maybe it is worth adding them into the plots with some discussions.

- 45  $\rightarrow$  For Case 1 the COBALD backscatter instrument does not show any indication for particles in the lower stratosphere and thus is a sign that all ice particles have evaporated by that time. For Case 2, unfortunately no COBALD measurements are available. As the COBALD data does not provide any important additional information it was not added to the paper for simplicity.
- L327: I suggest emphasizing here that the thickness of the humid layer from the observation (vertical dimension in m or km?) is much thinner than the resolution of the MLS data in the lower stratosphere which is in the scale of kilometers

 $\rightarrow$  The vertical spread of the water vapor peak is  $\approx$ 800 m for Case 1 and  $\approx$ 600 m for Case 2. We added the sentence: 'While the vertical extent of the water vapor peak is 800 m (600 m for Case 2), the vertical resolution of MLS  $H_2O$  is 1.5 km.'

55 - L485-486 I suggest also to point out the need in the future satellite mission that in high resolution in the vertical dimension is crucial for understanding the water vapor distribution in the UTLS

 $\rightarrow$  Changed last sentence to: 'Due to the low resolution of satellite data, in-situ measurements and future satellite missions with very high vertical resolution in the UTLS are therefore required to understand the impact of such small-scale events like overshooting convection.'

60 Technical corrections:

**- L81 is there any altitude dependency of the humidity sensor?**

 $\rightarrow$  Yes, we know that the accuracy of the RS41 radiosonde humidity sensor decreases with altitude due to the low temperatures. Here, we used the official values provided by the manufacturer in: "Survo Petteri, Raisa Lehtinen, Jari Kauranen: SI traceability of vaisala radiosonde RS41 sounding data - calibration and uncertainty analysis, pp. 6–9, 2008". It was shown that the RS41 has a higher uncertainty at lower temperature within 2-3% RH at -80C. Due to this issue we use, when possible, two humidity sensors i.e. the CFH as the second and more reliable humidity sensor. We added a sentence: "Petteri et al. 2008 shows a temperature dependency of the humidity sensor uncertainty, which does not exceed 3% RH at temperatures below -80C and RH below 30%."

**- Figure 2 I suggest to write H2O instead of H2O in the x and y labels. Same for the other figures (and O3 as well).**

 $\rightarrow$  Corrected.

65

70

75

80

85

- L178-180 **Please verify the numbers used in this phrase** "Between pressure levels of 180 hPa and 162.5 hPa, which correspond to potential temperature levels of 345 K and 357.5 K, the water vapor mixing ratio fluctuates between 5 ppmv and 7.4 ppmv and between 6 ppmv and 14.5 ppmv as measured by the radiosonde and the CFH respectively, before it attains the stratospheric background value of  $\approx$  4-5 ppmv, which is reached within all Case 1 profiles below the 160 hPa/360 K level. A background value of  $\approx$ 5 ppmv agrees well with results of previous studies (?)."
  - $\rightarrow$  More exact numbers inserted.
- Figure 3 In figure 3, I see the values are mostly around 4 ppmv instead of  $5 \rightarrow$  Values of around 4 ppmv are indeed reached shortly before the water vapor peak at 162.5 hPa. However, before that at pressure levels of 180-162.5 hPa (and the corresponding potential temperature levels), the water vapor mixing ratios vary between 5 and 7.4 ppmv. We corrected the sentence to 'Between pressure levels of 180 hPa and 162.5 hPa, which correspond to potential temperature levels of 345 K and 357.5 K, the water vapor mixing ratio fluctuates between 5 ppmv and 7.4 ppmv and between 6 ppmv and 14.5 ppmv as measured by the radiosonde and the CFH respectively, before it attains the stratospheric background value of  $\approx$  4-5 ppmv, which is reached within all Case 1 profiles below the 160 hPa/360 K level.'

**- Figure 3: repeating phrase in the caption: 'The gray region marks the level between 365K and 370K in which the water vapor enhancement is observed'**

 $\rightarrow$  Both phrases combined to : "The gray regions mark the level between 145 hPa and 165 hPa in a) and between 365 K and 370 K in b) in which the water vapor enhancement is observed."

**RC 2** 2**

**90**

95

**Regarding the relative levels of the water vapor and ozone peaks: Are sampling time constants for each instrument accounted for? If the response time for water/CFH is slower than that for the ECC cell, I can imagine a case where their peak locations might be offset?**

 $\rightarrow$  Indeed, we didn't correct for a time lag of the ECC ozone instrument. Thus, the ozone data were reevaluated with the new data processing method described by Vömel et al., 2020. With this method one can directly see the drop inside of the ozone peak in both plots, using pressure and potential temperature as a vertical coordinate, which coincides with the water vapor peak. So both profiles fit even better to each other (see Figure 1). Figures 3, 5 and 9 in the manuscipt are updated, accordingly.

---

## Editor Decision (ED1)

**Technical corrections ACP-2021-749 Revised**

P1, L4: radiation → radiative

P1, L15-16: "….ERA5 overestimates water vapor mixing ratios". Do you mean: "….the by ERA5 overestimated water vapor mixing ratios." (if yes, I would also suggest to write it like this)

P1, L19: Be more precise. What does it mean when the brightness temperature is low?

P2, L37: Space between "and" and reference "Wang and Huang (2020) missing.

P2, L40: particle → particles

P2, L47: in previous work → "in a previous work" or "in previous works"

P2, L53: simulation → simulations (or "a simulation")

P3, L54: simulation → simulations

P3, L58: stratospere → stratosphere

P3, L59: within a MOSES → within the MOSES

P3, L60: investigate extreme events → investigate extreme weather events

P3, L64: add why is there rarely the opportunity for such studies?

P3, L70: section → Section

P3, L72: section 5 → Section 5 (capital letters and remember, these are written full length at the begin of the sentence, otherwise abbreviated to Sect.)

P3, L73: move "using backward trajectories and satellite data" behind in Section 3.5.

P3, L74, Section 2 header: methods → method

P4, L95:  Change to "An ECC (Electrochemical …….) instrument (Smit et al., 2007)…..

P4, L105: appendix A → Appendix A

P4, L106: in mid-latitudes → in the mid-latitudes

P4, L107: appendix A2 → Appendix A2 and move "(see Appendix A2)" at the end of the sentence.

P4, L109: times → time

P4, L115: flights without the ….. → flights that were performed without the……

P5, L122: are → were

P6, L132: version 4 → Version 4 (?)

P7, L141: space missing between "where" and "1" and number and unit.

P7, L145: p is pressure → p is the pressure

P7, L147: potential temperature only shows → potential temperature shows only

P7, L147: and "thus", so that it reads "and can thus be considered….."

P8, L187: write "ranging" instead of "range" or write "with a temperature range"

P8, L190: either delete "pressure level" or write "is at a pressure level of ……"

P8, L202: in A3 → in Appendix A3

P8, Fig 4 caption: Add "that", so that it reads ….. (Case 2) that passed the measurement site…..

P11, L253: can not → cannot

P11, L258: remove "obtained by the authors" (obsolete)

P12, Fig 5 caption, 3$^{rd}$ line: show → shows and move "ascent and descent" after shows so that it reads "the ascent and descent data…..". Also delete here "by the authors" and write "for the data obtained from 2018 and 2002" instead of solely "from 2018 to 2020" and add "the" so that it reads "in the mid-latitudes".

P13, L278: ...and 8 → and Fig. 8

P14, L322: along towards → write either "along" or "towards"

P14, Sec. 3.5 header: Add "the", so that it reads "….along the CLaMS trajectories"?

P15, Fig. 7 caption: space between "ERA5" and "(a-c)" missing

P17, L333: appear → appears

P17, L348: H2O in upright font

P17, L360: this is supported by that fact…... → this is supported by the fact that…..

P19, Fig 10 caption: space between full stop and "The time…." missing.

P19, Fig 10 caption: text misleading (what is shown by which line) → rephrase. The time is shown as vertical blue line and with the red symbols.

P19, L374: Panel a in Figure 11 → Figure 11a

P19, L383: Same here.

P19, L388: seems → seem

P20, L389: pass near convective events → pass by near convective events

P20, L389: at a greater distance from what?

P20, L394: that the water vapor → that the observed water vapor

P21, L397: add the dates

P21, L398: from gravity breaking -> from gravity wave braking

P22, L411: in Figure 6 → in Fig. 6

P22, L411: I would write here " …as provided (or given) in Fig. 6 shows…….."

P22, L428: as well as ERA5 → as well as by ERA5

P23, L448: move "into the LS" after "in the" and replace "in" by "of" so that it reads "…..due to the very humid conditions of the into the LS mixed-in tropospheric airmass". Stop the sentence here and continue then with "Further, ……" and delete "which".

P23, L456: Why sparse??

P23, L474: It should read "Overall, the ERA5 data indicates……"

P24, L483: measurements → measurement

P24, L506: move "thoroughly" after "to" so that is reads "is crucial to thoroughly understand and quantify….."

P25, L528: Reference missing (shown with "?")

P28, Figure A2 caption: "F" after the full stop obsolete